SciPost Physics

Submission

# Soliton confinement in the double sine-Gordon model

S. B. Rutkevich

Fakultät für Mathematik und Naturwissenschaften, Bergische Universität Wuppertal,
42097 Wuppertal, Germany

## Abstract

The double sine-Gordon field theory in the weak confinement regime is studied. It represents the small non-integrable deformation of the standard sine-Gordon model caused by the cosine perturbation with the frequency reduced by the factor of 2. This perturbation leads to the confinement of the sine-Gordon solitons, which become coupled into the 'meson' bound states. We classify the meson states in the weak confinement regime, and obtain three asymptotic expansions for their masses, which can be used in different regions of the model parameters. It is shown, that the sine-Gordon breathers, slightly deformed by the perturbation term, transform into the mesons upon increase of the sine-Gordon coupling constant.

## 1   Introduction

Integrable models of statistical mechanics and quantum field theory [1,2] play an important role in the present understanding of the properties of the condense matter systems in the critical region close to the point of the continuous phase transition. Due to the universality of critical fluctuations, the exact solution of such a model provides a mathematically

reliable information not only about the integrable model itself, but also about the whole universality class it represents.

It turn out, however, that a number of interesting physical phenomena, such as particle decay, nucleation of domains of the equilibrium phase in the false vacuum surrounding, confinement of topological excitations, etc., require non-integrable models for their description. If the proper non-integrable model can be viewed as a small deformation of some integrable one, perturbative expansions around the integrable point could provide a useful insight into the realm of physical phenomena described by such a non-integrable model [2, 3]. This idea was first realized in the form factor perturbation theory (FFPT), which was introduced by Delfino, Mussardo, and Simonetti [4].

In this paper we study the non-integrable deformation of the sine-Gordon field theory, which is known in the literature as the two-frequency or double sine-Gordon (dsG) model [5–9]. We address the particular case of this model, which is specified by the Euclidean action:

$$\mathcal{A}_{dsG} = \int d^2 x \left[ \frac{1}{8\pi} (\partial_\mathfrak{a} \varphi)^2 + V(\varphi) \right], \tag{1}$$

where $\mathfrak{a} = 0, 1$, and

$$V(\varphi) = -2\mu \cos(\beta\varphi) - 2\Lambda \cos(\beta\varphi/2). \tag{2}$$

The parameters $\mu, \beta$, and $\Lambda$ are positive, and $0 < \beta \le 1$. Along with the sine-Gordon coupling constant $\beta$, the so-called renormalized coupling constant $\xi \in (0, +\infty)$, which is related with $\beta$ as

$$\xi = \frac{\beta^2}{1 - \beta^2}, \tag{3}$$

is also widely used.

At $\Lambda = 0$, action (1) describes the standard integrable sine-Gordon (sG) model. The description of its various properties can be found, *e.g.* in the monograph [2] and references therein. At $\Lambda > 0$, the second term in the right-hand side of (2) breaks the model integrability. It was shown by Delfino and Mussardo [6], that this perturbation term induces a linear attractive potential between the solitons of the sG model leading to their confinement: isolated solitons do not exist anymore in the system, and two solitons bind into compound particles - the 'mesons'. Recently, the $\Lambda$-dependencies of the mass of the lightest meson at several fixed values of the parameter $\beta$ in model (1), (2) were studied numerically by Roy and Lukyanov [10].

The aim of the present work is to classify the meson states in the dsG model (1) in the weak confinement regime at small $\Lambda \to +0$, and to calculate the meson mass spectra perturbatively in this small parameter. We obtain several initial terms of three asymptotic expansions, which describe the meson masses for $\xi \in (0, +\infty)$. It is shown, in particular, that at any small $\Lambda > 0$, there is no qualitative difference between the sG breathers, which survive at $\Lambda = 0$ for $\xi < 1$, and the newly formed mesons, which are present only at $\Lambda > 0$. At a fixed small $\Lambda > 0$, the breather, slightly deformed by the perturbation $\sim \Lambda$, smoothly transforms upon increase of the parameter $\xi$ into the newly formed meson state.

The rest of the paper is organised as follows. In Section 2 we remind some well-known properties of the sine-Gordon field theory and its two-frequency deformation defined by (1), (2). Section 3 contains a brief review of two perturbative techniques, which were recently developed for the calculation of the meson mass spectra in several QFT and spin-chain models in the weak confinement regime. Applying one of these techniques to the dsG model (1) at a small $\Lambda > 0$, we obtain two asymptotic expansions for the meson masses, which hold in different regions of the meson parameters. In Section 4 we derive the third asymptotic expansion, which determines the masses of light mesons for $\xi$ in the crossover intervals close to the inverse natural numbers $\xi_n = 1/n$. Together, three

asymptotic formulas obtained in Sections 3 and 4 determine the evolution of masses of all meson modes upon tuning the parameter $\xi$ in the whole range of its variation $0 < \xi < \infty$. This evolution is clarified and illustrated in Section 5. Concluding remarks are presented in Section 6. Finally, Appendix A contains the calculation of the meson energy spectra in the deformed Lieb-Liniger model. The results of this appendix are used in Section 4.

## 2 Sine-Gordon model and its deformation

In this section we remind some well-known facts about the pure SG model and its deformation (1), (2).

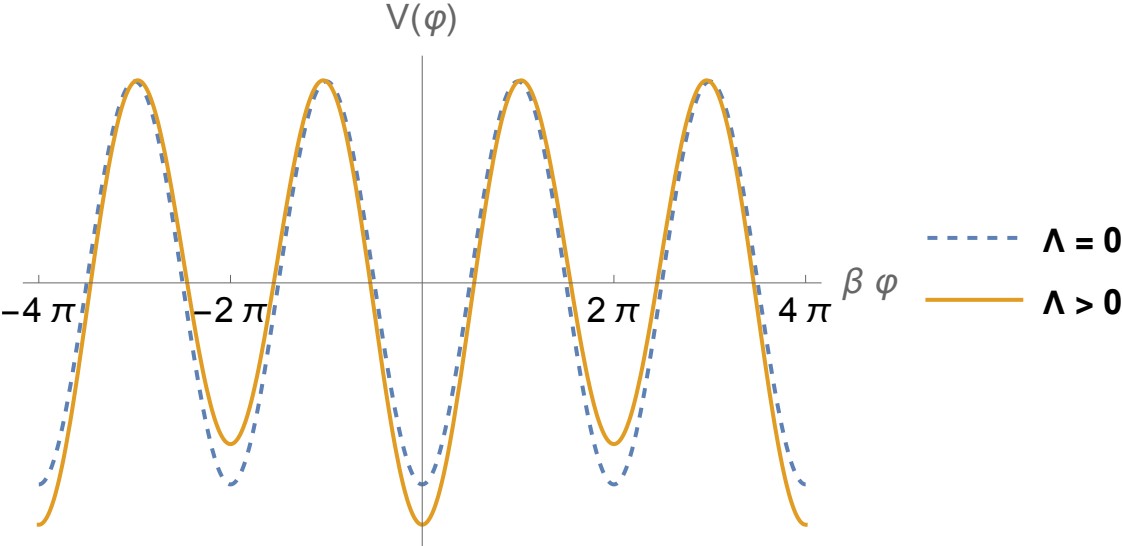

Figure 1: The potential (2) at $\Lambda = 0$ (dashed curve), and at a small $\Lambda > 0$ (solid curve).

The profiles of the potential $V(\varphi)$ at $\Lambda = 0$, and at a small $\Lambda > 0$ are shown in Figure 1 by the dashed, and solid curves, respectively. At $\Lambda = 0$, the potential (2) is the $2\pi/\beta$-periodical function of $\varphi$, which takes its minimum value

$$V_{min}(\mu, \Lambda = 0) = -2\mu, \tag{4}$$

at $\varphi = 2\pi l/\beta$, with $l \in \mathbb{Z}$. At $\Lambda \neq 0$, the period of the potential $V(\varphi)$ increases by the factor of 2. At a small enough $\Lambda \in (0, 1/4)$, the locations of minima remain the same, $\varphi = 2\pi l/\beta$, while the minimum values of the potential $V(\varphi)$ corresponding to even and odd $l$ become shifted downwards, and upwards, respectively:

$$V_{min}(\mu, \Lambda, l) = \begin{cases} -2\mu - 2\Lambda, & \text{for even } l, \\ -2\mu + 2\Lambda, & \text{for odd } l. \end{cases} \tag{5}$$

Due to the $4\pi/\beta$-periodicity of the potential (2), it is natural to compactify the scalar field variable $\varphi$ by putting it on the circle with the circumference $4\pi/\beta$:

$$\varphi \sim \varphi + \frac{4\pi l}{\beta}, \text{with } l \in \mathbb{Z}.$$

Upon this identification, the model (1) reduces at $\Lambda = 0$ to the two-folded sine-Gordon model $sG(\beta, 2)$ [11]. In the infinite volume, this model has two energetically degenerate

vacuums $|vac\rangle^{(a)}$, $a = 0, 1$, with the energy density $\mathcal{E}_{sG}$. In the *repulsive regime* at $\xi > 1$, the particle sector is represented solely by the kinks (solitons and antisolitons) $|K_{ab}(\alpha)\rangle_s$ interpolating between the vacua $a$ and $b$. In the *attractive regime* at $\xi \in (0, 1)$, the two-kink bound states (the breathers) are also allowed at $\Lambda = 0$ [12]. The kink state $|K_{ab}(\alpha)\rangle_s$ is characterized by the rapidity $\alpha \in \mathbb{R}$, and by the isospin (topological charge) $s$, which takes the value $+1/2$ for solitons, and $-1/2$ for antisolitons. The energy $\omega$ and the momentum $p$ of a kink can be parametrized by its rapidity $\alpha$ in the usual way:

$$\omega(\alpha) = m \cosh \alpha, \quad p(\alpha) = m \sinh \alpha, \tag{6}$$

where $m$ is the kink mass. The dimensional coupling constant $\mu$ in the sG potential (2) is related to the kink mass $m$ as follows:

$$\mu = \kappa(\xi) \, m^{2/(\xi+1)}, \tag{7}$$

where $\xi$ is the renormalized coupling constant (3), and the constant $\kappa(\xi)$ is known due to Al. B. Zamolodchikov [13]:

$$\kappa(\xi) = \frac{1}{\pi} \frac{\Gamma\left(\frac{\xi}{\xi+1}\right)}{\Gamma\left(\frac{1}{\xi+1}\right)} \left[\frac{\sqrt{\pi}\, \Gamma\left(\frac{\xi+1}{2}\right)}{2\, \Gamma\left(\frac{\xi}{2}\right)}\right]^{2/(\xi+1)}. \tag{8}$$

In the attractive regime at $\xi \in \left(\frac{1}{j+1}, \frac{1}{j}\right)$, there are $j$ breather states with masses [12,14]:

$$m_j^{(b)}(\xi) = 2m \sin\left(\frac{\pi j \xi}{2}\right). \tag{9}$$

We will distinguish the breathers with odd and even $j$ by the breather parity:

$$\iota(j) = (-1)^j. \tag{10}$$

The variation of the breather masses with $\xi$ for five lightest breather is shown in Figure 2. The continuous spectrum in the two-soliton sector, which is shown in grey in this figure, fills the energies $E > 2m$.

The two-kink excitations in the two-folded sine-Gordon model sG$(\beta, 2)$ are represented by the states $|K_{ab}(\alpha_1)K_{ba}(\alpha_2)\rangle_{s_1 s_2}$. We shall use also the two-kink states $|K_{ab}(\alpha_1)K_{ba}(\alpha_2)\rangle_{\pm}$ defined by the relations:

$$|K_{ab}(\alpha_1)K_{ba}(\alpha_2)\rangle_{\pm} = \frac{1}{\sqrt{2}}[|K_{ab}(\alpha_1)K_{ba}(\alpha_2)\rangle_{1/2, -1/2} \tag{11}$$
$$\pm |K_{ab}(\alpha_1)K_{ba}(\alpha_2)\rangle_{-1/2, 1/2}].$$

The two-kink scattering matrix in the sG model was found by A. B. Zamolodchikov [12]. Its representation in the sG$(\beta, 2)$ model can be described by the commutation relations:

$$|K_{ab}(\alpha_1)K_{ba}(\alpha_2)\rangle_{1/2, 1/2} = S_0(\alpha_1 - \alpha_2)|K_{ab}(\alpha_2)K_{ba}(\alpha_1)\rangle_{1/2, 1/2}, \tag{12a}$$

$$|K_{ab}(\alpha_1)K_{ba}(\alpha_2)\rangle_{-1/2, -1/2} = S_0(\alpha_1 - \alpha_2)|K_{ab}(\alpha_2)K_{ba}(\alpha_1)\rangle_{-1/2, -1/2}, \tag{12b}$$

$$|K_{ab}(\alpha_1)K_{ba}(\alpha_2)\rangle_{\pm} = S_{\pm}(\alpha_1 - \alpha_2)|K_{ab}(\alpha_2)K_{ba}(\alpha_1)\rangle_{\pm}, \tag{12c}$$

where

$$S_{\iota}(\alpha, \xi) = -e^{i\theta_{\iota}(\alpha, \xi)}, \tag{13}$$

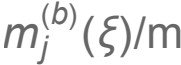

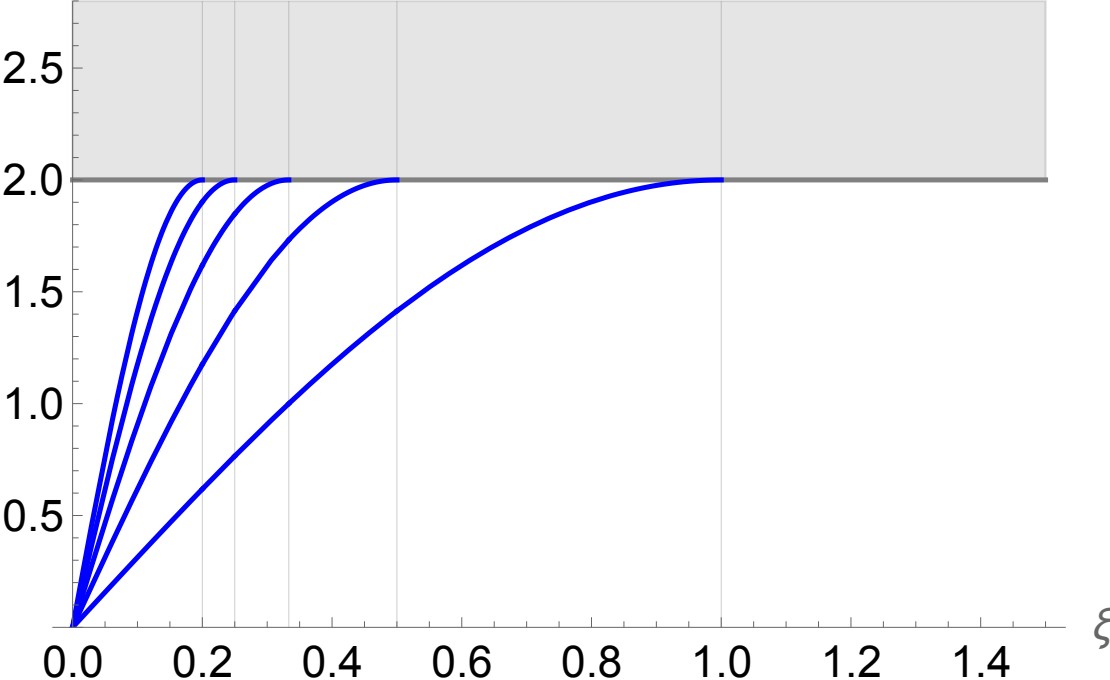

Figure 2: The $\xi$-dependencies due to (9) of the masses of five lightest breathers.

with $\iota = 0, \pm$. The scattering phase $\theta_0(\alpha)$ is given by the integral formula:

$$\theta_0(\alpha, \xi) = -\int_0^\infty \frac{dx}{x} \frac{\sinh\left(\frac{x}{2\xi} - \frac{x}{2}\right) \sin\left(\frac{\alpha x}{\pi \xi}\right)}{\sinh(x/2) \cosh\left(\frac{x}{2\xi}\right)}. \tag{14}$$

The scattering phases $\theta_\pm(\alpha, \xi)$ are determined by formulas:

$$\theta_\pm(\alpha, \xi) = \chi_\pm(\alpha) + \theta_0(\alpha, \xi), \tag{15a}$$

$$\chi_+(\alpha, \xi) = \arg\left[\frac{\sinh[(i\pi + \alpha)/(2\xi)]}{\sinh[(i\pi - \alpha)/(2\xi)]}\right], \tag{15b}$$

$$\chi_-(\alpha, \xi) = \arg\left[\frac{\cosh[(i\pi + \alpha)/(2\xi)]}{\cosh[(i\pi - \alpha)/(2\xi)]}\right]. \tag{15c}$$

Note, that the soliton-antisoliton scattering amplitudes $S_\pm(\alpha, \xi)$ degenerate at integer $\xi^{-1}$. Namely,

$$S_+(\alpha, \xi) = -S_0(\alpha, \xi), \quad S_-(\alpha, \xi) = S_0(\alpha, \xi), \text{ for } \xi^{-1} = 2l, \tag{16}$$

$$S_+(\alpha, \xi) = S_0(\alpha, \xi), \quad S_-(\alpha, \xi) = -S_0(\alpha, \xi), \text{ for } \xi^{-1} = 2l - 1, \tag{17}$$

with natural $l = 1, 2 \ldots$. In particular, in the free-fermion point $\xi = 1$:

$$S_+(\alpha, 1) = S_0(\alpha, 1) = -1, \quad S_-(\alpha, 1) = 1. \tag{18}$$

Perturbation of the two-folded sine-Gordon model by the term $-2\Lambda \cos(\beta\varphi/2)$ with a small $\Lambda > 0$ in the potential (2) breaks the degeneracy between the vacua $|vac\rangle^{(a)}$, $a = 0, 1$. The vacuum $|vac\rangle^{(1)}$ associated with potential minima at even $l$ in equation (4) decreases

in energy and becomes the true ground state, while the vacuum $|vac\rangle^{(0)}$ associated with minima with odd $l$ increases in energy and transforms into the metastable false vacuum. To the linear oder in $\Lambda \to +0$, the energy densities of the deformed vacua are given [10] by the straightforward perturbative formulas:

$$\mathcal{E}_{psG}^{(1)}(\Lambda) = \mathcal{E}_{sG} - 2\Lambda\,\mathcal{G}_{\beta/2} + O(\Lambda^2), \tag{19}$$

$$\mathcal{E}_{psG}^{(0)}(\Lambda) = \mathcal{E}_{sG} + 2\Lambda\,\mathcal{G}_{\beta/2} + O(\Lambda^2), \tag{20}$$

where $\mathcal{G}_\gamma$ is the vacuum expectation value of the exponential operator $e^{\pm i\gamma\varphi}$ at $\Lambda = 0$. The explicit expression for this expectation value was found by Lukyanov and Zamolodchikov [15]:

$$\mathcal{G}_\gamma = \left[\pi\mu\,\frac{\Gamma(1-\beta^2)}{\Gamma(\beta^2)}\right]^{\frac{\gamma^2}{1-\beta^2}} \tag{21}$$

$$\times \exp\left(\int_0^\infty \frac{dt}{t}\left(\frac{\sinh^2(2\gamma\beta t)}{2\sinh(\beta^2 t)\sinh(t)\cosh\left((1-\beta^2)t\right)} - 2\gamma^2\,e^{-2t}\right)\right).$$

The margin between the energies of the false and true vacua induces the linear attractive potential $v(x_1, x_2) = \mathfrak{f}\cdot(x_2 - x_1)$ between two kinks located near the points $x_1$ and $x_2$, such that $x_2 - x_1 \gg m^{-1}$. Here $\mathfrak{f}$ denotes the *string tension*:

$$\mathfrak{f} = \mathcal{E}_{psG}^{(0)}(\Lambda) - \mathcal{E}_{psG}^{(1)}(\Lambda) = f + O(\Lambda^3), \tag{22}$$

where $f = 4\Lambda\,\mathcal{G}_{\beta/2}$. In what follows, we will widely use instead of the parameter $f$ its dimensionless counterpart

$$\lambda = \frac{f}{m^2}. \tag{23}$$

The linear attractive potential leads to confinement of kinks, which become coupled into the 'meson' bound states [8,16,17]. The meson states $|\pi_{s,\iota,n}(P)\rangle$ can be classified by the momentum $P$, isospin $s = 0, \pm 1$, parity $\iota = 0, \pm$, and the quantum number $n = 1, 2, 3 \ldots$. The quantum numbers $s$ and $\iota$ are not independent: $\iota = 0$ for $s = \pm 1$, and $\iota = \pm$ at $s = 0$.

At $\Lambda = 0$, the meson state $|\pi_{s,\iota,n}(P)\rangle$ decouples into some linear combinations of the two-kink states:

$$|\pi_{s=1,\iota=0,n}(P)\rangle \to |K_{10}(\alpha_1)K_{01}(\alpha_2)\rangle_{1/2,1/2}, \tag{24}$$

$$|\pi_{s=-1,\iota=0,n}(P)\rangle \to |K_{10}(\alpha_1)K_{01}(\alpha_2)\rangle_{-1/2,-1/2}, \tag{25}$$

$$|\pi_{s=0,\iota=\pm,n}(P)\rangle \to |K_{10}(\alpha_1)K_{01}(\alpha_2)\rangle_\pm, \tag{26}$$

with $p(\alpha_1) + p(\alpha_2) = P$. So, the meson modes with the isospin value $s = 1$ ($s = -1$) represent the bound states of two solitons (antisolitons), while the two meson modes with $s = 0$ represent the soliton-antisoliton bound states with different parities $\iota = \pm$.

The mesons $|\pi_{s,\iota,n}(P)\rangle$ are relativistic particles with the dispersion law:

$$E_{\iota,n}(P) = \sqrt{M_{\iota,n}^2 + P^2}, \tag{27}$$

where $M_{\iota,n}$ are the meson masses.

Since the perturbed sine-Gordon model (1) is not integrable, the analytic calculation of the meson masses $M_{\iota,n}$ is possible only by means of some perturbative technique. In this paper we focus on the weak confinement regime of the perturbed sine-Gordon model (1), (2), in which the constant $\Lambda > 0$ is treated as a small parameter of the theory.

# 3 Weak coupling expansions for the meson masses

At the first sight, it would be natural to use the FFPT for studying the confinement phenomenon in the non-integrable deformations of integrable QFTs. It was realized, however, already in [4, 6] that the original version of the FFPT introduced in [4] cannot be applied directly to the confinement problem, since the confinement transition changes the particle content of the theory. In order to circumvent this problem, two different perturbative techniques for calculation of the meson masses in the weak confinement regime were developed.

The first more systematic and sophisticated technique exploits the Bethe-Salpeter equation. The main advantage of this approach is that it allows one to account the long-range attraction between confined particles already in the zero order of the perturbation theory. In the high-energy physics, the Bethe-Salpeter equation was applied to the confinement problem in the two-dimensional QCD with infinite number of colours by t'Hooft [18]. Fonseca and Zamolodchikov [16, 17] derived the Bethe-Salpeter equation for the Ising field theory (IFT) and applied it to calculation of the meson masses in this model in the limit of a weak magnetic field. Later the technique based on the Bethe-Salpeter equation has been used for calculation of the meson energy spectra in three other (properly deformed) models exhibiting the kink confinement: in the quantum Ising spin chain [19], in the gapped antiferromagnetic XXZ spin-1/2 chain [20], and in the three-state Potts field theory [21].

It was showed by means of this technique, that the meson masses $M_n(\lambda)$ in the IFT in the weak confinement regime can be described by two asymptotic weak-coupling expansions in the dimensionless small parameter $\lambda \to +0$, which is defined by equation (23).

(i) The *low-energy expansion* in fractional powers of $\lambda$ describes the meson masses $M_n(\lambda)$ slightly above the edge point $2m$, i.e. at $n \ll \lambda^{-1}$ and $\lambda \to +0$. Many terms in this expansion were obtained by Fonseca and Zamolodchikov [17]. Two leading ones read:

$$M_n(\lambda) = 2m + \lambda^{2/3}\, m\, z_n + O(\lambda^{4/3}), \tag{28}$$

where $-z_n$ are the zeroes of the Airy function, $\mathrm{Ai}(-z_n) = 0$. These two terms of the IFT low-energy expansion reproduce the old result of McCoy and Wu [22].

(ii) The *semiclassical expansion* [17, 23] in integer powers of $\lambda$ describes the masses of highly excited meson states with $n \gg 1$. It is convenient to parametrise the masses of such mesons by the set of rapidities $\alpha_n$:

$$M_n(\lambda) = 2m \cosh \alpha_n. \tag{29}$$

These rapidities must solve the transcendent equation with the power series in $\lambda$ in the right-hand side. To the leading order in $\lambda$, this equation reads:

$$\sinh(2\alpha_n) - 2\alpha_n = \lambda[2\pi(n - 1/4)] + O(\lambda^2). \tag{30}$$

The second perturbative technique is not so powerful and rigorous, but, instead, rather heuristic and intuitive. Nevertheless, it allows one in a very simple way to recover the initial parts of the weak coupling expansions for the meson masses beyond the order $O(\lambda^2)$. This technique can be viewed as a generalisation of the phenomenological scenario of confinement, which was introduced by McCoy and Wu [22] in 1978.

Proceeding to the description of this heuristic technique, let us consider a system of two relativistic particles of the mass $m$ moving in a line and attracting one another with a linear potential. Their Hamiltonian reads:

$$H(x_1, x_2, p_1, p_2) = \sqrt{p_1^2 + m^2} + \sqrt{p_2^2 + m^2} + f|x_1 - x_2|, \tag{31}$$

where $x_1, x_2 \in \mathbb{R}$, and $f > 0$. Since the total momentum of two particles $P = p_1 + p_2$ is conserved, one can concentrate on their dynamics for the states with zero total momentum $P = 0$. Then, the relative motion of two particles is determined by the reduced Hamiltonian:

$$\widetilde{H}(x, p) = 2\sqrt{p^2 + m^2} + f|x|, \tag{32}$$

where $x = x_1 - x_2 \in \mathbb{R}$. The variables $x$ and $p$ in (32) represent the canonical coordinate and momentum. The classical trajectories of the system (32) are formed by periodically repeated cycles. Each cycle starts from the collision between two particles, and then follows by their subsequent movement with a constant acceleration.

Let us now turn to the quantization of the simple classical dynamics determined by the Hamiltonian (32). The quantum version of the system (32) has a discrete energy spectrum, and the energies of the two-particle bound states can be identified with the meson masses $M_n$, with $n = 1, 2, \ldots$.

For $n \gg 1$, the masses can be obtained by means of the semiclassical quantization. Exploiting parametrization (29), one obtains in the straightforward fashion [17, 23] the constraint on the rapidities $\alpha_n$ following from the Bohr-Sommerfeld quantization rule. The result depends on the particle statistics. If two particles are fermions, the Bohr-Sommerfeld quantization condition leads to the equation:

$$\sinh(2\alpha_n) - 2\alpha_n = \lambda[2\pi(n - 1/4)], \tag{33}$$

in agreement with (30). In the case of two bosons, one gets, instead:

$$\sinh(2\alpha_n) - 2\alpha_n = \lambda[2\pi(n - 3/4)]. \tag{34}$$

A different quantization scheme should be applied for calculation of the masses $M_n(\lambda)$ of light mesons, such that

$$\frac{M_n(\lambda)}{2m} - 1 \ll 1.$$

This strong inequality implies, that the momentum $p$ in the reduced Hamiltonian (32) is small $|p| \ll m$, and, therefore, one can replace this Hamiltonian by its non-relativistic approximation:

$$\widetilde{H}(x, p) = 2m + \frac{p^2}{m} + f|x| + O(p^4). \tag{35}$$

After canonical quantization $p \to -i\partial_x$, the reduced Hamiltonian (35) transforms into the Schrödinger Hamiltonian operator

$$\hat{H} = 2m - \frac{\partial_x^2}{m} + f|x|. \tag{36}$$

The set of its eigenvalues again depends on the particle statistics. If the two particles are fermions, the Hamiltonian operator (36) acts in the space of odd wave functions, $\psi(-x) = -\psi(x)$, and the meson masses are given by formula (28). In the case of two bosons, the wave function must be even, $\psi(-x) = \psi(x)$, and the meson masses are, instead, determined by the equation

$$M_n(\lambda) = 2m + \lambda^{2/3} m z'_n + O(\lambda^{4/3}), \tag{37}$$

where $-z'_n$ denote the zeroes of the derivative of the Airy function,

$$\mathrm{Ai}'(z)\big|_{z=-z'_n} = 0.$$

In summary, two asymptotic expansions describing the meson masses in the IFT in the weak confinement regime can be derived by means of the systematic perturbative technique based on the Bethe-Salpeter equation. Few initial terms in these expansions can be recovered in a very simple way by means of the quantization (semiclassical or canonical) of the classical dynamics of two particles, which is determined by the Hamiltonian (31).

It is important to note, that the IFT has a very specific property: the kink topological excitations do not interact in the deconfined phase at zero magnetic field behaving like free fermions. In contrast, in many other integrable models, including the sine-Gordon field theory, elementary excitations strongly interact already in the deconfined phase at short distances, and this interaction is encoded in the non-trivial scattering matrix. Fortunately, the outlined above heuristic perturbative technique, upon minor modifications, can be also applied to such models. This modified heuristic technique was first introduced for the calculation of the meson masses in the Potts field theory [24], and later [20, 25] applied to calculation of the meson energy spectra in the antiferromagnetic XXZ spin chain perturbed by the staggered longitudinal magnetic field. In paper [20], this technique was described in much detail, and results for the meson energy spectra obtained by this method were confirmed by more rigorous calculations exploiting the perturbative solution of the Bethe-Salpeter equation. In the rest of this section, we will discuss very briefly this modified heuristic technique. More details can be found in [20].

In integrable QFT, the short range interaction between particles is described by the Faddeev-Zamolodchikov commutation relation, which contains the elastic two-particle scattering matrix. Since the latter can be diagonalised by means of an appropriate unitary transform, it is sufficient to concentrate on the case of the one-channel scattering. In this case, the Faddeev-Zamolodchikov commutation relation reduces to the form:

$$Z^*(\alpha_1)Z^*(\alpha_2) = S(\alpha_1 - \alpha_2)Z^*(\alpha_2)Z^*(\alpha_1), \tag{38}$$

where $Z^*(\alpha)$ is the operator creating the particle with rapidity $\alpha$, and $S(\alpha)$ is the scattering amplitude. It follows from the unitarity condition

$$S(\alpha)S(-\alpha) = 1, \tag{39}$$

that the scattering amplitude can be written in the form:

$$S(\alpha) = -e^{i\theta(\alpha)}, \tag{40}$$

where $\theta(\alpha)$ is the scattering phase. We will require, that this scattering phase is an odd function of the rapidity: $\theta(-\alpha) = -\theta(\alpha)$.

The non-trivial two-particle scattering modifies the semiclassical quantization of the classical dynamics determined by the Hamiltonian (32): the scattering phase of two colliding particles must be added to the left-hand side of the Bohr-Sommerfeld condition, as it was explained in [20, 24]. As the result, the semiclassical formulas (29), (33) for the meson masses modify to the form (cf. eqs. (171), (172) in [20]):

$$M_n(\lambda) = 2m\cosh\alpha_n, \tag{41}$$
$$\sinh(2\alpha_n) - 2\alpha_n = \lambda[2\pi(n - 1/4) - \theta(2\alpha_n)], \tag{42}$$

with $n \gg 1$.

It follows from (39), that $S(0)^2 = 1$. Accordingly, one should distinguish the fermionic $S(0) = -1$, and bosonic $S(0) = 1$ cases. In the fermionic case, the scattering phase $\theta(\alpha)$ is continuous at $\alpha \in \mathbb{R}$, and $\theta(0) = 0$. In the bosonic case, the scattering phase $\theta(\alpha)$ must have a discontinuity at the origin, and one can set: $\lim_{\alpha \to \pm 0} \theta(\alpha) = \pm \pi$. In the latter case, it is convenient to introduce the 'bosonic' scattering phase $\tilde{\theta}(\alpha) = \theta(\alpha) - \pi \operatorname{sign} \alpha$, which is odd in $\alpha$, and continuous at the origin: $\tilde{\theta}(0) = 0$. Equation (42) can be then rewritten in terms of the 'bosonic' scattering phase $\tilde{\theta}(\alpha)$:

$$\sinh(2\alpha_n) - 2\alpha_n = \lambda[2\pi(n - 3/4) - \tilde{\theta}(2\alpha_n)]. \tag{43}$$

In the case of free bosons $\tilde{\theta}(\alpha) \equiv 0$, the latter equation reduces to (34).

The short-range interaction between particles in the deconfined phase effects also the low energy expansion for the meson masses in the weak confinement regime. In order to account this effect in the frame of the heuristic approach outlined above, one should add some short-range interaction potential $u(x_1 - x_2)$ in the classical phenomenological Hamiltonian (31). After the canonical quantization, the potential $u(x)$ emerges in the right-hand side of the Schrödinger Hamiltonian operator (36). It was shown in [20] in the fermionic case $S(0) = -1$, that this leads to the additional term $\sim \lambda$ in the low-energy expansion (28) for the meson masses (see equation (187) in [20]):

$$M_n(\lambda) = 2m + \lambda^{2/3} m z_n + \lambda m^2 a + O(\lambda^{4/3}), \tag{44}$$

where $a$ denotes the scattering length [26], which is defined as:

$$a = -m^{-1} \partial_\alpha \theta(\alpha) \Big|_{\alpha = +0}. \tag{45}$$

The same term $\sim \lambda$ appears as well in the low-energy expansion (37) in the bosonic case $S(0) = 1$:

$$M_n(\lambda) = 2m + \lambda^{2/3} m z'_n + \lambda m^2 a + O(\lambda^{4/3}). \tag{46}$$

It is important to note, that the derivation of formula (44) described in [20] was essentially based on the assumption, that the scattering phase $\theta(\alpha)$ smoothly depends on $\alpha$ at $|\alpha| \lesssim 1$. Now let us apply the general results described above to the dsG model (1). In the *weak confinement regime* at $\Lambda \to +0$, the meson masses in the dsG model admit two different asymptotic expansions in the dimensionless parameter $\lambda = f/m^2$.

(i) At large $n \gg 1$ and small $\lambda \ll 1$, the meson masses can be described by the *semiclassical* asymptotic expansion in integer powers of $\lambda$. The meson masses in this semiclassical regime are determined by the formula

$$M_{\iota,n}(\lambda, \xi) = 2m \cosh \alpha_{\iota,n}, \tag{47}$$

in terms of the rapidities $\alpha_{\iota,n}$, which solve the transcendent equation

$$\sinh(2\alpha_{\iota,n}) - 2\alpha_{\iota,n} = \lambda[2\pi(n - 1/4) - \theta_\iota(2\alpha_{\iota,n}, \xi)] + O(\lambda^2), \tag{48}$$

and the two-kink scattering phases $\theta_\iota(\alpha, \xi)$ are given by equations (14), (15).

(ii) At a fixed $n \sim 1$ and $\lambda \to +0$ the *low-energy* asymptotic expansion in fractional powers of $\lambda$ holds. Three leading terms of this expansion read:

$$M_{\iota,n}(\lambda, \xi) = 2m + \lambda^{2/3} m z_n + \lambda m^2 a_\iota(\xi) + O(\lambda^{4/3}), \tag{49}$$

where $-z_n$ are the zeroes of the Airy function, $\operatorname{Ai}(-z_n) = 0$, and $a_\iota(\xi)$ denote the scattering length:

$$a_\iota(\xi) = -m^{-1} \partial_\alpha \theta_\iota(\alpha, \xi) \Big|_{\alpha = 0}. \tag{50}$$

In the explicit form, the scattering lengths $a_\iota(\xi)$ in the sine-Gordon model read:

$$a_0(\xi) = -\frac{1}{2\pi m \xi} \int_{-\infty}^{\infty} dx \left[ 1 - \tanh \frac{x}{2\xi} \coth \frac{x}{2} \right], \tag{51}$$

$$a_+(\xi) = a_0(\xi) + \frac{1}{m\xi} \operatorname{ctg} \frac{\pi}{2\xi}, \tag{52}$$

$$a_-(\xi) = a_0(\xi) - \frac{1}{m\xi} \tan \frac{\pi}{2\xi}. \tag{53}$$

In the deformed sine-Gordon model, the asymptotic formulas (48), (49) hold at generic values of the parameter $\xi$, which are not too close to the inverse natural numbers $1, 1/2, 1/3, \ldots$. Indeed, the soliton-antisoliton scattering lengths $a_+(\xi)$ and $a_-(\xi)$ given by formulas (52), and (53) diverge, respectively, at $\xi^{-1} = 2l$, and at $\xi^{-1} = 2l - 1$, with natural $l$. Directly at these exceptional values of the parameter $\xi$, formulas (52), and (53) must be modified in the following way.

At $\xi = \frac{1}{2l}$, the semiclassical expansion of the masses $M_{\iota,n}(\lambda, \beta)$ of the soliton-antisoliton bound states with the positive parity $\iota = +$ is still described by equation (47) with large integer $n$, in which the rapidities $\alpha_{+,n}$ now solve the equation:

$$\sinh(2\alpha_{+,n}) - 2\alpha_{+,n} = \lambda[2\pi(n - 3/4) - \theta_0(2\alpha_{+,n}, \xi)] + O(\lambda^2). \tag{54}$$

In turn, the low-energy expansion for the masses $M_{+,n}(\lambda, \beta)$ at $\xi = \frac{1}{2l}$ takes the form

$$M_{+,n}(\lambda, \xi) = 2m + \lambda^{2/3} m \, z_n' + \lambda \, m^2 \, a_0(\xi) + O(\lambda^{4/3}), \tag{55}$$

where $-z_n'$ denotes the zeroes of the derivative of the Airy function. At $\xi = \frac{1}{2l-1}$, with $l = 1, 2, \ldots$, the semiclassical and low energy expansions for the masses of the soliton-antisoliton bound states with the negative parity $\iota = -$ are modified in the similar way:

$$\sinh(2\alpha_{-,n}) - 2\alpha_{-,n} = \lambda[2\pi(n - 3/4) - \theta_0(2\alpha_{-,n}, \xi)] + O(\lambda^2), \tag{56}$$

$$M_{-,n}(\lambda, \xi) = 2m + \lambda^{2/3} m \, z_n' + \lambda \, m^2 \, a_0(\xi) + O(\lambda^{4/3}). \tag{57}$$

# 4 Low-energy expansions at $\xi^{-1}$ close to natural numbers

At $\xi^{-1}$ close to the natural numbers, the low energy expansion (49) for the meson masses $M_{\pm,n}(\lambda, \xi)$ must be further modified. In this section, we will obtain the modified low-energy expansion for the meson masses $M_{-,n}(\lambda, \xi)$, which will replace (49) (for $\iota = -$) in the crossover regions of the parameter $\xi$ close to $\xi_{2l-1} = \frac{1}{2l-1}$, i.e. for

$$\xi = \frac{1}{2l - 1} + \delta\xi, \text{ with } |\delta\xi| \ll 1, \tag{58}$$

where $l = 1, 2 \ldots$. The similar modified low-energy expansion for the meson masses $M_{+,n}(\lambda, \xi)$ at $\xi = \frac{1}{2l} + \delta\xi$, with natural $l$ and small $|\delta\xi|$, will be presented by the end of the section.

At $\xi = \xi_{2l-1}$, the two-kink scattering phases $\theta_+(\alpha, \xi)$ and $\theta_0(\alpha, \xi)$ coincide:

$$\theta_+(\alpha, \xi_{2l-1}) = \theta_0(\alpha, \xi_{2l-1}). \tag{59}$$

This odd function smoothly depends on the rapidity $\alpha$, and at small $|\alpha| \ll 1$ behaves as:

$$\theta_0(\alpha, \xi_{2l-1}) = \alpha \, \partial_\alpha \theta_0(\alpha, \xi_{2l-1})|_{\alpha=0} + O(\alpha^3). \tag{60}$$

Tuning the parameter $\xi$ from the point $\xi_{2l-1}$ by a small $\delta\xi$ induces only the uniformly small in $\alpha$ variations $\delta\theta_0(\alpha, \xi)$ of this scattering phase: $|\delta\theta_0(\alpha, \xi)| < C|\delta\xi|$, where the positive number $C$ does not depend on $\alpha$.

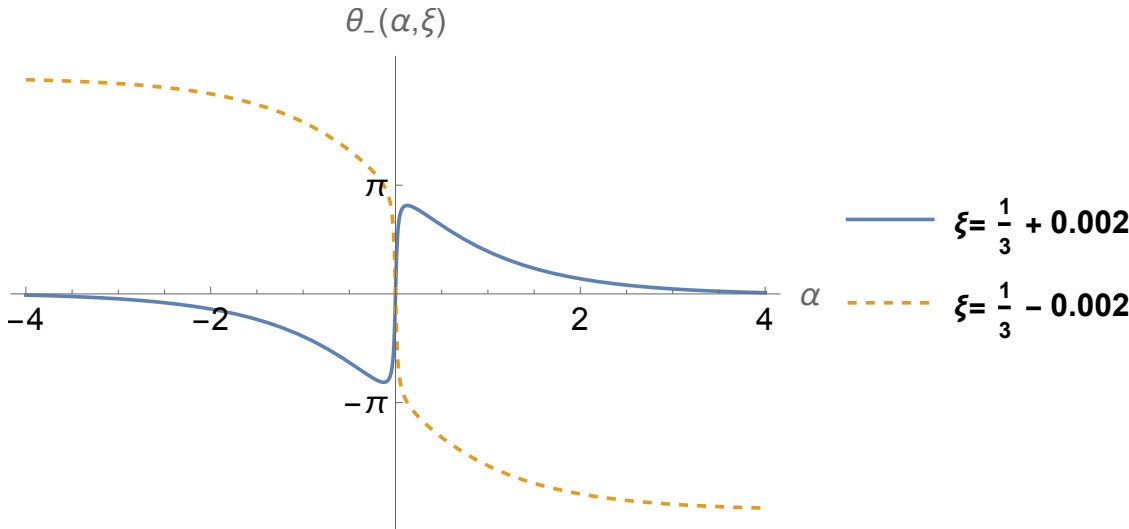

Figure 3: Plot of the scattering phase $\theta_-(\alpha, \xi)$ determined by (15a) at $\xi = \frac{1}{3} + 0.002$, and at $\xi = \frac{1}{3} - 0.002$.

In contrast, the scattering phase $\theta_-(\alpha, \xi)$ at $\xi$ in the crossover region (58) displays a fast variation at small real $\alpha$. For two values of $\xi$ close to $1/3$, this fast variation of the scattering phase $\theta_-(\alpha, \xi)$ near the origin is illustrated in Figure 3.

It was noticed in Section 3, that a smooth variation of the scattering phase $\theta(\alpha)$ at small $\alpha$ was the essential requirement exploited in the derivation of the low-energy expansion (44). Since this requirement is broken for the scattering phase $\theta_-(\alpha, \xi)$ at $\xi$ in the crossover region (58), the low-energy expansion (49) for the masses $M_{-,n}(\lambda, \xi)$ of light mesons cannot be used there. We proceed now to the derivation of the different low-energy asymptotic formula for the meson masses $M_{-,n}(\lambda, \xi)$, which will replace (49) in the crossover regions (58).

First, let us note, that the fast variation of the scattering phase $\theta_-(\alpha, \xi)$ at $\xi$ close to $\xi_{2l-1}$ is induced, mathematically, by approaching to the real $\alpha$-axis of the pole and zero of the scattering amplitude $S_-(\alpha, \xi)$, which lie in the imaginary $\alpha$-axis at the points $\alpha_B = -i\pi(2l-1)\delta\xi$, and $-\alpha_B$, respectively. At $\delta\xi < 0$, the pole $\alpha_B$ enters the physical sheet $0 < \operatorname{Im}\alpha < \pi$, indicating appearance of the $(2l-1)$th breather in the theory at $\xi < \xi_{2l-1}$.

Taking into account the above mentioned analytical properties of the the scattering amplitude $S_-(\alpha, \xi)$, we rewrite the latter for $\xi$ in the vicinity of the point $\xi_{2l-1}$ in the form

$$S_-(\alpha, \xi_{2l-1} + \delta\xi) = -S_-^{(reg)}(\alpha, \xi_{2l-1} + \delta\xi)\frac{\alpha - i\pi(2l-1)\,\delta\xi}{\alpha + i\pi(2l-1)\,\delta\xi}. \tag{61}$$

The function $S_-^{(reg)}(\alpha, \xi_{2l-1} + \delta\xi)$ in the right-hand side has at $|\delta\xi| \ll \xi_{2l-1} - \xi_{2l+1}$ neither poles nor zeroes in the narrow strip $|\operatorname{Im}\alpha| < \gamma_l$ with $\gamma_l \approx 2\pi\xi_{2l-1}$, and

$$S_-^{(reg)}(0, \xi_{2l-1} + \delta\xi) = -1. \tag{62}$$

At small $\alpha$, the kink rapidity becomes proportional to its momentum:

$$\alpha(p) = \frac{p}{m} + O(p^3), \tag{63}$$

and the kink relativistic dispersion law (6) can be approximated by its non-relativistic asymptotics:

$$\omega(p) = m + \frac{p^2}{2m} + O(p^4). \tag{64}$$

As it was shown by McCoy and Wu [22], the low-energy expansions for the meson masses in relativistic QFTs in the weak confinement regime can be effectively studied in the frame of the non-relativistic quantum mechanics. Following this strategy, let us first consider a system of two non-relativistic $\delta$-interacting bosons moving in a line. The Hamiltonian of the system reads:

$$\mathcal{H}_0 = 2m - \frac{\partial^2_{x_1}}{2m} - \frac{\partial^2_{x_2}}{2m} + c\,\delta(x_1 - x_2), \tag{65}$$

where $c$ is the interaction constant.

The quantum state of two bosons can be described by the wave-function $\Psi(x_1, x_2)$, with $x_1, x_2 \in \mathbb{R}$, which is symmetric under permutation of the spatial coordinates $x_1, x_2$. Up to the constant term $2m$, the Hamiltonian (65) coincides with the Hamiltonian of the Lieb-Liniger model in the two-particle sector, which is discussed in Appendix A.

The solution of the Hamiltonian eigenvalue problem

$$\mathcal{H}_0\,\Psi_{p_1,p_2}(x_1, x_1) = E(p_1, p_2)\,\Psi_{p_1,p_2}(x_1, x_1) \tag{66}$$

in the half-plane $-\infty < x_1 < x_2 < +\infty$ is given by the Bethe wave function:

$$\Psi_{p_1,p_2}(x_1, x_1) = \exp[i(p_1 x_1 + p_2 x_2)] + \mathcal{S}(p_1 - p_2)\exp[i(p_1 x_2 + p_2 x_1)], \tag{67}$$

where

$$\mathcal{S}(p_1 - p_2) = \frac{p_1 - p_2 - ic\,m}{p_1 - p_2 + ic\,m} \tag{68}$$

is the two-particle scattering amplitude, and

$$E(p_1, p_2) = 2m + \frac{p_1^2}{2m} + \frac{p_1^2}{2m} \tag{69}$$

is the energy of the state (67).

Upon the choice

$$c = (2l - 1)\pi\,\delta\xi, \tag{70}$$

the scattering amplitude (68) coincides with the leading term of the small-$\alpha$ asymptotics of the scattering amplitude $S_-(\alpha_1 - \alpha_2, \xi_{2l-1} + \delta\xi)$ of the sine-Gordon model

$$\mathcal{S}(p_1 - p_2) = S_-(\alpha_1 - \alpha_2, \xi_{2l-1} + \delta\xi)\Big|_{\alpha_{1,2}=p_{1,2}/m} + O(p_1 - p_2). \tag{71}$$

Let us now deform the Hamiltonian (65) by adding the linear attractive potential acting between bosons:

$$\mathcal{H}(f) = 2m - \frac{\partial^2_{x_1}}{2m} - \frac{\partial^2_{x_2}}{2m} + c\,\delta(x_1 - x_2) + f|x_1 - x_2|, \tag{72}$$

where $f > 0$ is the string tension. The energy spectrum $\{E_n(f, c)\}_{n=1}^{\infty}$ of the particle bound states with zero total momentum can be easily found, as it is described in Appendix A. The final result reads:

$$E_n = 2m + m\lambda^{2/3}t_n, \tag{73}$$

where $\lambda = f/m^2$, and the numbers $t_n$ are the successive solutions of the transcendent equation

$$Y_\rho(t_n) = 0, \tag{74}$$

with $Y_\rho(t) = \mathrm{Ai}'(-t) - \rho\,\mathrm{Ai}(-t)$. Here $\mathrm{Ai}(z)$ is the Airy function, and

$$\rho = \frac{c}{2\lambda^{1/3}}. \tag{75}$$

Formulas (73)-(75), and (70) determine two leading terms in the modified low-energy expansion for the masses of mesons having the negative parity $\iota = -$ in the crossover region (58). One more term $\sim \lambda$ in this low-energy expansion can be determined following the lines described in [20] by taking into account the smooth dependence of the function $S_-^{(reg)}(\alpha, \xi)$ on the rapidity $\alpha$ at $\alpha \to 0$. Then, the modified low-energy expansion representing in the crossover region (58) the masses of mesons with parity $\iota = -$ takes the form:

$$M_{-,n}(\lambda, \xi_{2l-1} + \delta\xi) = 2m + m\lambda^{2/3}t_n + \lambda\, m^2\, a_0(\xi_{2l-1}) + O(\lambda^{4/3}), \tag{76}$$

where the scattering length $a_0(\xi)$ is given by (50).

Let us describe the meson masses predicted by equation (76) in three limit cases.

(i) At $\rho \gg 1$, the solutions of the secular equation (74) have the following asymptotics

$$t_n(\rho) = z_n - \rho^{-1} + O(\rho^{-2}).$$

Accordingly, formula (76) reduces at $\lambda^{1/3} \ll \xi - \xi_{2l-1} \ll 1$ to the form

$$M_{-,n}(\lambda, \xi)$$

$$= 2m + m\lambda^{2/3}\left(z_n - \frac{2\lambda^{1/3}}{c} + \ldots\right) + \lambda\, m^2\, a_0(\xi_{2l-1}) + O(\lambda^{4/3})$$

$$= 2m + m\lambda^{2/3}z_n + \lambda\, m^2\left[-\frac{2}{cm} + a_0(\xi) + \ldots\right] + O(\lambda^{4/3})$$

$$= 2m + m\lambda^{2/3}z_n + \lambda\, m^2\, a_-(\xi) + O(\lambda^{4/3}),$$

in agreement with (49).

(ii) At $\rho = 0$, solutions of equation (74) taken with the 'minus' sign are just the zeroes of the derivative of the Airy function: $t_n(0) = z_n'$, $\mathrm{Ai}'(-z_n') = 0$, and formula (76) reduces to (57).

(iii) At a large negative $\rho$, the first solution $t_1(\rho)$ of equation (74) becomes negative and well separated from all other solutions $t_2(\rho), t_3(\rho), \ldots$. The following asymptotic formulas hold at $\rho \to -\infty$:

$$t_1(\rho) = -\rho^2 - \frac{1}{2\rho} + O(\rho^{-4}), \tag{77}$$

and for $n = 2, 3 \ldots$:

$$t_n(\rho) = z_{n-1} - \rho^{-1} + O(\rho^{-2}).$$

As the result, we obtain at $\lambda^{1/3} \ll -\delta\xi \ll 1$ for the meson masses (76):

$$M_{-,1}(\lambda, \xi_{2l-1} + \delta\xi) = 2m - \frac{mc^2}{4} \tag{78}$$

$$+ \lambda m\left[-\frac{1}{c} + m\, a_0(\xi_{2l-1}) + O(\delta\xi)\right] + \ldots$$

$$= 2m - \frac{\pi^2 m}{4}\left(\frac{\delta\xi}{\xi_{2l-1}}\right)^2 + \lambda m\left[m\, a_-(\xi_{2l-1} + \delta\xi) + \frac{\xi_{2l-1}}{\pi\delta\xi}\right] + \ldots$$

and

$$M_{-,n}(\lambda, \xi_{2l-1} + \delta\xi) = 2m + m\lambda^{2/3}z_{n-1} + \lambda\, m^2\, a_-(\xi_{2l-1} + \delta\xi) + O(\lambda^{4/3}), \quad (79)$$

for $n = 2, 3, \ldots$.

The lightest meson with the mass given by (78) must be identified with the $j$th breather, with $j = 2l - 1$, perturbed by the term $-2\Lambda\cos(\beta\varphi/2)$ in the potential (2). Indeed, the mass $m^{(b)}_{2l-1}(\xi)$ of this breather at $f = 0$ and $\xi < \xi_{2l-1}$ is given by equation (9):

$$m^{(b)}_{2l-1}(\xi) = 2m\sin\left(\frac{\pi\xi}{2\,\xi_{2l-1}}\right).$$

Two initial terms in its Taylor expansion at $\xi = \xi_{2l-1}$ read:

$$m^{(b)}_{2l-1}(\xi) = 2m - \frac{\pi^2 m}{4}\left(\frac{\xi - \xi_{2l-1}}{\xi_{2l-1}}\right)^2 + O\big((\xi - \xi_{2l-1})^4\big). \quad (80)$$

At $\lambda = 0$, the last line of (78) reduces to the right-hand side of (80), in confirmation of the above statement. The linear in $\lambda$ term in the last line of (78) represents the linear perturbation of the breather energy by the deforming potential $-2\Lambda\cos(\beta\varphi/2)$.

The three initial terms of the modified low-energy expansion for the masses of the mesons with $\iota = +$ at $\xi$ close to $(2l)^{-1}$ can be obtained by means of the same procedure. The only difference is that formula (70) must be now replaces by:

$$c = 2l\,\pi\,\delta\xi.$$

The final result reads:

$$M_{+,n}(\lambda, \xi) = 2m + m\lambda^{2/3}t_n + \lambda\, m^2\, a_0(\xi_{2l}) + O(\lambda^{4/3}), \quad (81)$$

where $\xi = \xi_{2l} + \delta\xi$, $\xi_{2l} = (2l)^{-1}$, and $|\delta\xi| \ll 1$.

It follows from (81), that at

$$\lambda^{1/3} \ll \xi_{2l} - \xi \ll 1 \quad (82)$$

the masses of mesons with $\iota = +$ behave as:

$$M_{+,1}(\lambda, \xi_{2l} + \delta\xi) = 2m - \frac{\pi^2 m}{4}\left(\frac{\delta\xi}{\xi_{2l}}\right)^2 + \lambda\, m\left[m\, a_+(\xi_{2l} + \delta\xi) + \frac{\xi_{2l}}{\pi\delta\xi}\right] + \ldots, \quad (83)$$

and

$$M_{+,n}(\lambda, \xi_{2l} + \delta\xi) = 2m + m\lambda^{2/3}z_{n-1} + \lambda\, m^2\, a_+(\xi_{2l} + \delta\xi) + O(\lambda^{4/3}), \quad (84)$$

for $n = 2, 3, \ldots$. So, at $\xi$ satisfying (82), the lightest meson in this set is just the perturbed $(2l)$th breather.

So, upon decrease of the parameter $\xi$ in the crossover region close to the point $\xi_{2l-1} = 1/(2l-1)$, the lightest meson with the negative parity $\iota = -$ transforms into the $(2l-1)$th breather, which also has the negative parity according to (10). In turn, the lightest meson with the positive parity $\iota = +$ transforms upon decrease of $\xi$ close to the point $\xi_{2l} = 1/(2l)$ into the $(2l)$th breather, which parity is positive as well.

# 5   Evolution of meson masses with parameter $\xi$

In this section we illustrate and compare the predictions for the meson masses given by different asymptotic expansions, which were obtained in the previous section. The emphasis will be made on studying of the variation of these masses upon tuning the parameter $\xi$.

Figure 4 displays the masses $M_{\iota,n}(\lambda,\xi)$ of six lightest mesons determined by the semiclassical formulas (47), (48) at the fixed value of the parameter $\lambda = 0.5$ and five different values of the parameter $\xi$. Note, that the semiclassical approximation a priori is supposed to work well only for the mesons with large $n \gg 1$, while the masses of the lightest mesons should be instead described by the low-energy expansions. It will be shown later, however, that in many cases, the predictions for the masses of the lightest mesons (with small $n = 1, 2, \ldots$) given by the semiclassical and low-energy expansions are numerically very close to each other.*

The meson masses shown in Figure 4 form triplets, in which the mesons are distinguished by the parity index $\iota = 0, \pm$. The splitting of the meson masses inside each triplet is caused by the $\iota$-dependent scattering phase $\theta_\iota(2\alpha_{\iota,n}, \xi)$ in the right-hand sides of (48).

At $\xi > 1$, the meson masses in each triplet are ordered in the following way:

$$M_{-,n}(\lambda,\xi) < M_{0,n}(\lambda,\xi) < M_{+,n}(\lambda,\xi). \tag{85}$$

This situation is illustrated in Figures 4a, and 4b, which correspond to the $\xi$-parameter values $\xi = 2$, and $\xi = 1.2$, respectively.

Directly at the free-fermion point $\xi = 1$, two scattering phases vanish:

$$\lim_{\xi \to 1} \theta_\iota(\alpha, \xi) = 0, \quad \text{for } \iota = 0, +,$$

while the third one reduces to the step function:

$$\lim_{\xi \to 1 \pm 0} \theta_-(\alpha, \xi) = \pm \pi \, \mathrm{sign} \, \alpha. \tag{86}$$

Accordingly, the masses of the mesons with parities $\iota = 0, +$, continuously depend on $\xi$ near the free-fermion point $\xi = 1$, and $M_{0,n}(\lambda, 1) = M_{+,n}(\lambda, 1)$. In contrast, the masses of the mesons with negative parity display a discontinuity at $\xi = 1$, such that:

$$\lim_{\xi \to 1 - 0} M_{-,n}(\lambda,\xi) = \lim_{\xi \to 1 + 0} M_{-,n+1}(\lambda,\xi). \tag{87}$$

As the result, the masses of the mesons at $\xi \in (1/2, 1)$ in each triplet are ordered due to

$$M_{+,n}(\lambda,\xi) < M_{0,n}(\lambda,\xi) < M_{-,n}(\lambda,\xi), \tag{88}$$

as one can see in Figures 4c, and 4d.

The next reordering of the meson masses upon decrease of the parameter $\xi$ takes place at $\xi = 1/2$, and the order (85) is restored in the interval $\xi \in (1/3, 1/2)$, as it is shown in Figure 4d. Directly at $\xi = 1/2$ one gets from (47), (48), and (15): $M_{0,n}(\lambda, 1/2) = M_{-,n}(\lambda, 1/2)$, and

$$\lim_{\xi \to 1/2 - 0} M_{+,n}(\lambda,\xi) = \lim_{\xi \to 1/2 + 0} M_{+,n+1}(\lambda,\xi). \tag{89}$$

---

*The high efficiency of the semiclassical expansions for prediction the masses of lightest mesons in several models exhibiting confinement were noticed in papers [27–30], in which the meson masses were studied by direct numerical methods.

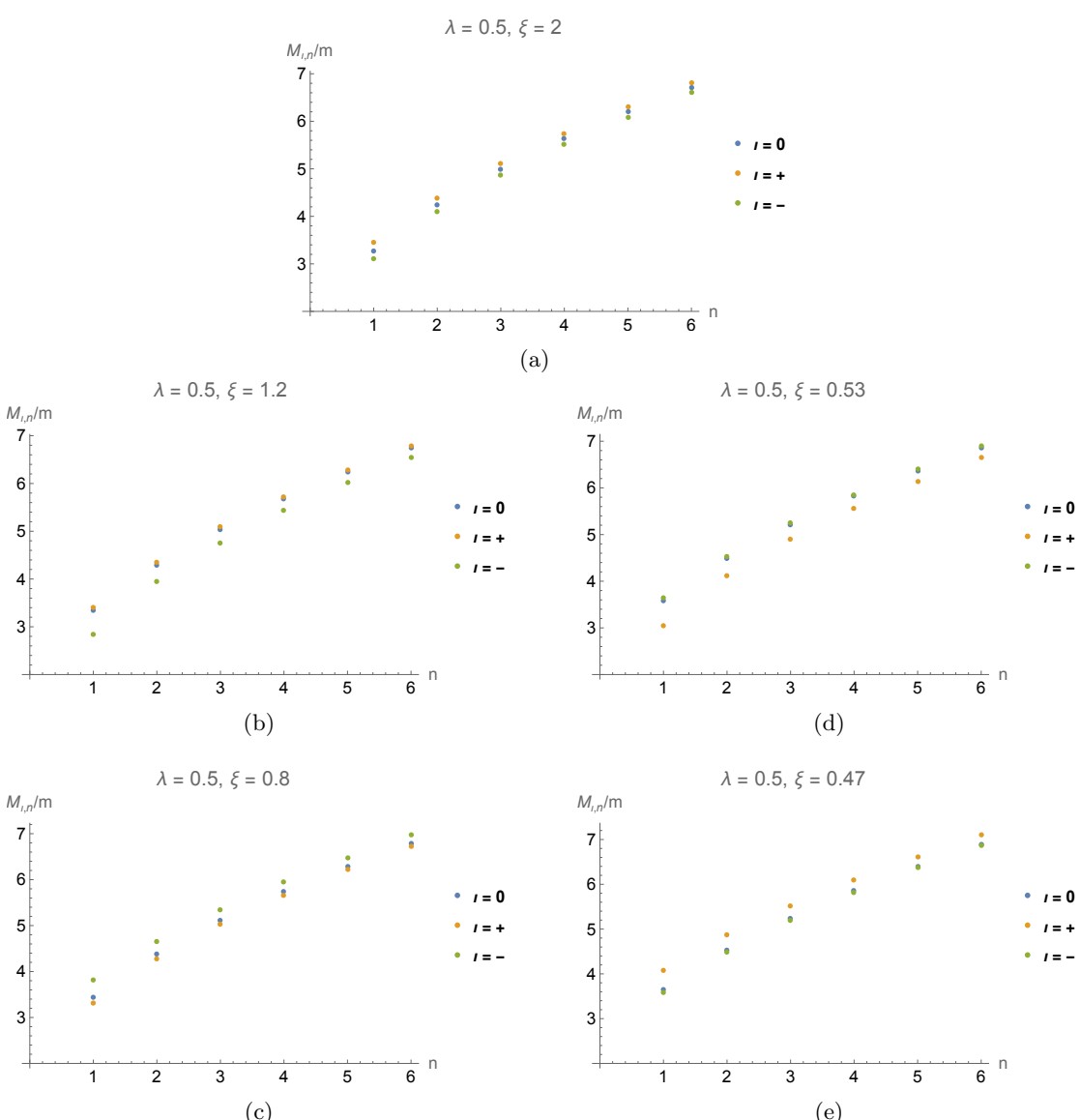

Figure 4: Masses of mesons $M_{\iota,n}(\lambda, \xi)$ calculated by semiclassical formulae (47), (48) at $\lambda = 0.5$, $\iota = 0, \pm$, and $n = 1, \ldots, 6$ for (a) $\xi = 2$, (b) $\xi = 1.2$, (c) $\xi = 0.8$, (d) $\xi = 0.53$, (e) $\xi = 0.47$.

In general, the meson masses in each triplet are ordered according to (85) at $\xi \in \left((2l+1)^{-1}, (2l)^{-1}\right)$, and due to (88) at $\xi \in \left((2l)^{-1}, (2l-1)^{-1}\right)$, with natural $l$.

Figure 5 displays the evolution of the masses of several lightest meson modes of different parities $\iota = 0, \pm$, calculated by means of the semiclassical formulas (47), (48), upon variation of the parameter $\xi$. The parameter $\lambda$ for all mesons in this figure is chosen at the fixed value $\lambda = 0.01$.

As one can see in Figure 5a, the masses of the mesons with the parity $\iota = 0$ monotonically increase with decreasing $\xi$. In contrast, the masses of the mesons with positive parity $\iota = +$, and fixed $n = 1, 2, \ldots$, shown in different colours in Figure 5b, monotonically decrease with decreasing $\xi$ in the intervals $\xi \in (1/2, \infty)$, $\xi \in (1/4, 1/2)$, $\xi \in (1/6, 1/4)$, $\ldots$, but have discontinuities at the points $\xi = 1/2, 1/4, 1/6, \ldots$. The masses of the mesons with negative parity $\iota = -$ shown in Figure 5c display the similar evolution upon variation of the parameter $\xi$, but with discontinuities at the points $\xi = 1, 1/3, 1/5, \ldots$.

It is clear, however, that the continuity in $\xi$ of the semiclassical meson mass spectra can be easily restored in both cases $\iota = \pm$ simply by changing the numbering of the meson states $M_{\iota,n}(\lambda, \xi)$:

$$\tilde{M}_{-,n}(\lambda, \xi) = M_{-,n-l}(\lambda, \xi), \qquad \text{at } \xi_{2l+1} < \xi < \xi_{2l-1}, \tag{90a}$$

$$\tilde{M}_{+,n}(\lambda, \xi) = M_{+,n-l}(\lambda, \xi), \qquad \text{at } \xi_{2l+2} < \xi < \xi_{2l}, \tag{90b}$$

where $\tilde{M}_{\iota,n}(\lambda, \xi)$ denotes the semiclassical meson masses with modified numbering, $l = 0, 1, \ldots, n-1$, $\xi_j = 1/j$ for natural $j$, and $\xi_0 = \xi_{-1} = +\infty$.

The masses of the meson modes $\tilde{M}_{\iota,n}(\lambda, \xi)$ defined by (90) are continuous (and analytical) functions of the real parameter $\xi$, which monotonically decrease with decreasing $\xi$. However, these functions are defined not for all positive $\xi$: $\tilde{M}_{-,n}(\lambda, \xi)$ and $\tilde{M}_{+,n}(\lambda, \xi)$ are defined in the half-infinite intervals $\xi \in (1/(2n-1), +\infty)$, and $\xi \in (1/(2n), +\infty)$, respectively. In fact, the positive left edges in these half-infinite intervals indicate the failure in the vicinity of these points of the semi-classical approximation used in the derivation of (48), (54). This statement is illustrated in Figure 6, which displays the evolution in $\xi$ close to the point $\xi_1 = 1$ of two lightest meson modes with the negative parity $\iota = -$ at the fixed value of $\lambda = 0.01$. The blue dots in this figure plot the $\xi$-dependencies the masses $\tilde{M}_{-,n}(0.01, \xi)$ with $n = 1, 2$, determined by the semiclassical formulas (90a), (47), (48). Note, that the explicit form of formula (90a) at $n = 1, 2$ reads:

$$\tilde{M}_{-,1}(\lambda, \xi) = M_{-,1}(\lambda, \xi), \qquad \text{at } 1 < \xi < \infty, \tag{91}$$

$$\tilde{M}_{-,2}(\lambda, \xi) = \begin{cases} M_{-,2}(\lambda, \xi), & \text{at } 1 < \xi < \infty, \\ M_{-,1}(\lambda, \xi), & \text{at } 1/3 < \xi < 1 \end{cases}. \tag{92}$$

A priori, however, there is no reason to rely on the semiclassical asymptotic formulas (47), (48) in the case of light mesons with $n = 1, 2$, since the semiclassical approximation can be well justified only for the highly excited meson states with $n \gg 1$. The masses of light mesons should be described, instead, by the low-energy expansions. The brown dotted-dashed curves in Figure 6 plot at $\xi > 1$ the masses $M_{-,n}(0.01, \xi)$ of two lightest mesons (with $n = 1, 2$) given by the low-energy expansion (49). As one can see from this figure, predictions of the low-energy and semiclassical expansions for these masses become numerically very close to each other at $\xi \gtrsim 1.4$. However, at $\xi$ in the crossover region close to the point $\xi_1 = 1$, the low-energy expansion (49) is not applicable, and one should use instead the modified low-energy expansion derived in the previous section. The solid red curves in Figure 6 show the predictions of this modified low-energy expansion (76) at $l = 1$ for the masses of two lightest mesons $M_{-,n}(0.01, \xi)$ with $n = 1, 2$ at $\xi$ in the crossover

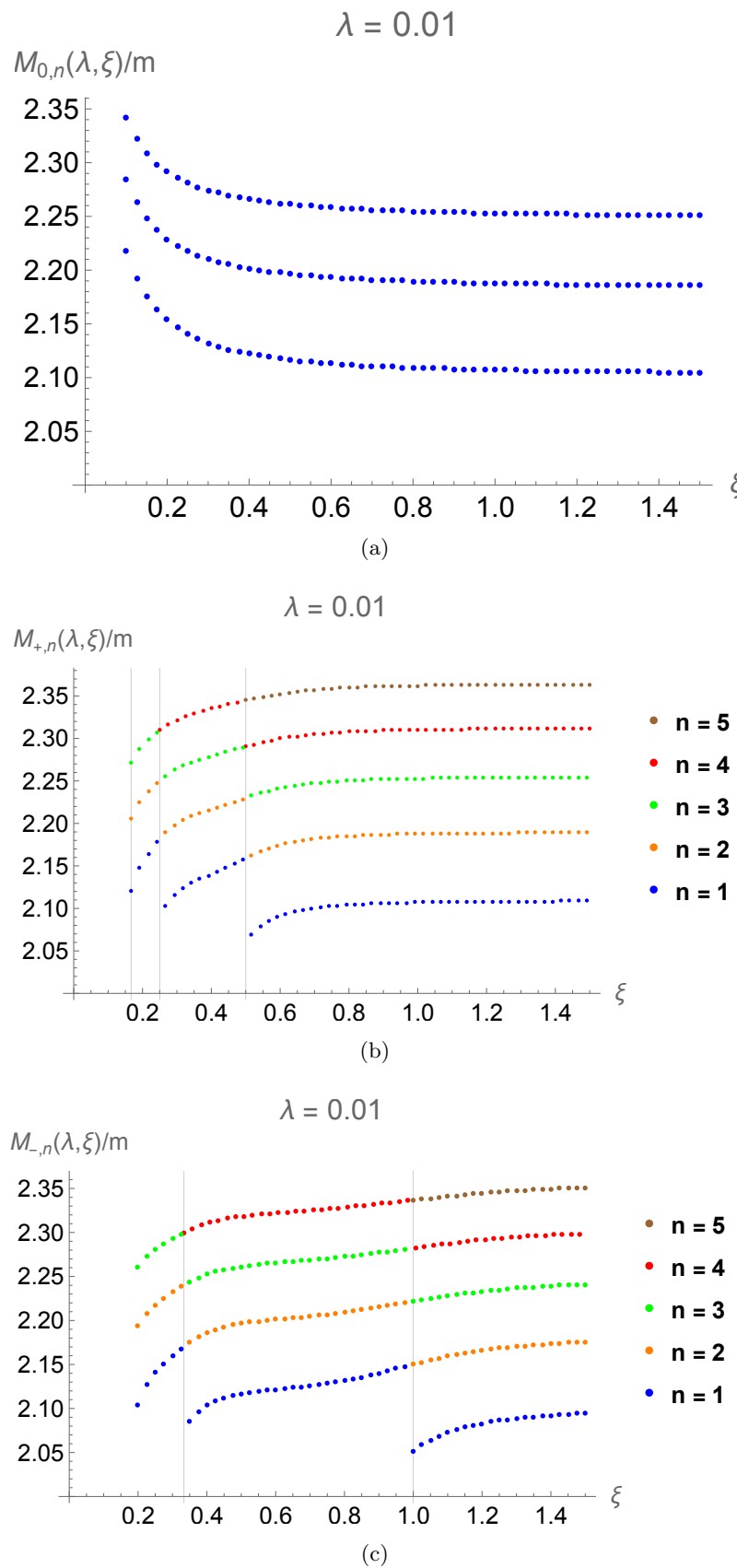

Figure 5: Variation with $\xi$ of the masses of lightest mesons $M_{\iota,n}(\xi)$ at $\lambda = 0.01$ calculated by semiclassical formulae (47), (48): (a) at $\iota = 0$, (b) at $\iota = +$, (c) at $\iota = -$ .

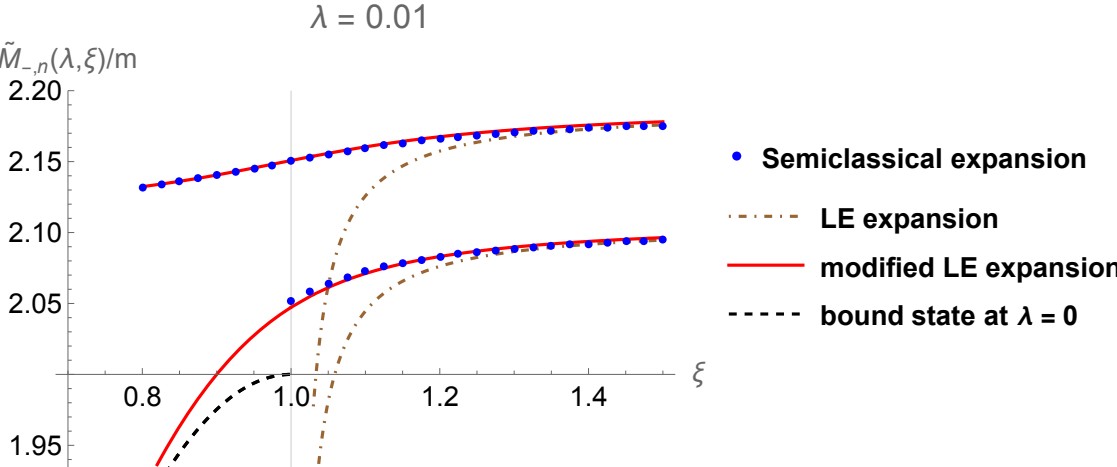

Figure 6: Variation with $\xi$ of the masses $\tilde{M}_{-,n}(\lambda,\xi)$ of two lightest meson modes with $n = 1, 2$ at $\lambda = 0.01$ and $\xi$ close to the free-fermion point $\xi_1 = 1$. The $\xi$-dependence of these masses given by the semiclassical expansion (47), (48), (90a) are shown by blue dots; the brown dotted-dashed lines display the variations with $\xi$ of these masses given by the low-energy expansion (49), (53); the solid red lines plot the prediction of the modified low-energy expansion (76) for $M_{-,n}(0.01,\xi)$ with $n = 1, 2$. The dashed black line shows the mass of the lightest breather $m_1^{(b)}(\xi)$ at $\lambda = 0$, which is given by (9).

region close to the point $\xi_1 = 1$. The mass $m_1^{(b)}(\xi)$ of the first breather at $\lambda = 0$ is plotted by the black dashed line in Figure 6.

Comparison of the upper blue dotted, and solid red curves in Figure 6 indicates, that the semiclassical formulas (47), (48), (92) give surprisingly accurate numerical values for the mass of the second meson mode in the wide interval of the parameter $\xi \in (0.8, 1.5)$ shown in this figure. In turn, the mass $M_{-,1}(0.01,\xi)$ of the lightest meson, which is determined by the modified low-energy expansion (76) and shown by the lower solid red curve in Figure 6, approaches the mass of the first breather $m_1^{(b)}(\xi)$ upon decrease of the parameter $\xi$ below the free-fermion point $\xi_1 = 1$. This indicates, that the lightest meson having the negative parity $\iota = -$ transforms upon decrease of the parameter $\xi$ below the free-fermion point into the breather, deformed by the perturbation term $-2\Lambda\cos(\beta\varphi/2)$ in the interaction potential (2).

It turns out, that all mesons characterized by the parities $\iota = \pm$ display qualitatively similar evolution upon decrease of the parameter $\xi$. Figures 7a, and 7b, show the $\xi$-dependencies of the masses of several meson modes with parities $\iota = +$, and $\iota = -$, respectively. As in Figures 5, 6, the parameter $\lambda$ is chosen at the fixed value $\lambda = 0.01$. The curves shown by the blue dots in Figure 7 display the semiclassical mass spectra determined by equations (47), (48). The solid red lines show the evolution of the masses $M_{\iota,1}(\lambda, \xi_l + \delta\xi)$ of the lightest mesons, which transform into the $l$th breather in the crossover regions surrounding the points $\xi_l = 1/l$. The masses $M_{\iota,1}(\lambda, \xi_l + \delta\xi)$ in the crossover regions are determined by the modified low-energy expansions (81), and (76) for the cases $\iota = +$, and $\iota = -$, respectively. The masses of the breathers $m_l^{(b)}(\xi)$ at $\lambda = 0$ are shown by the dashed black curves.

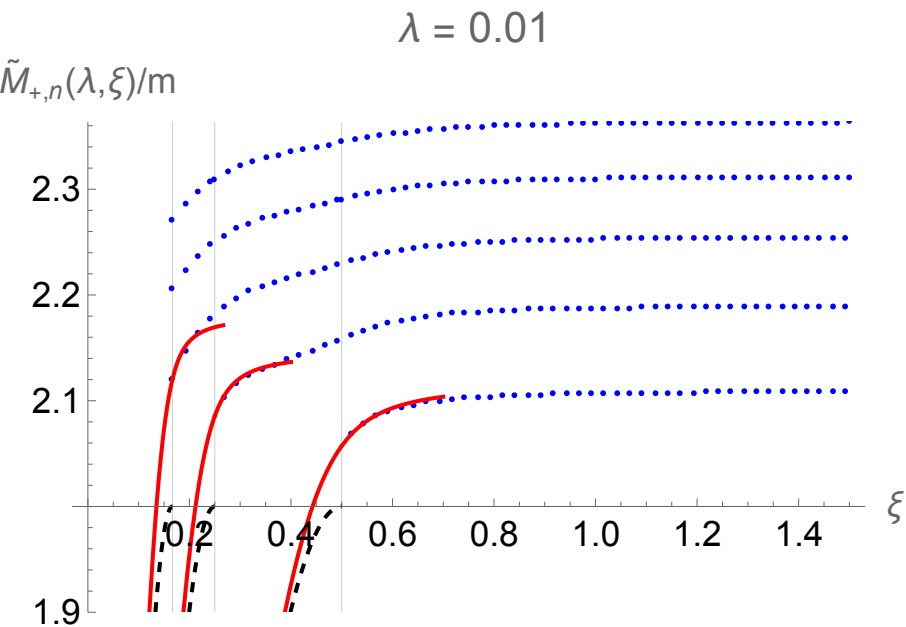

(a) Dashed black lines display the masses of breathers $m_j^{(b)}(\xi)$ with $j = 2, 4, 6$ in the unperturbed sine-Gordon model.

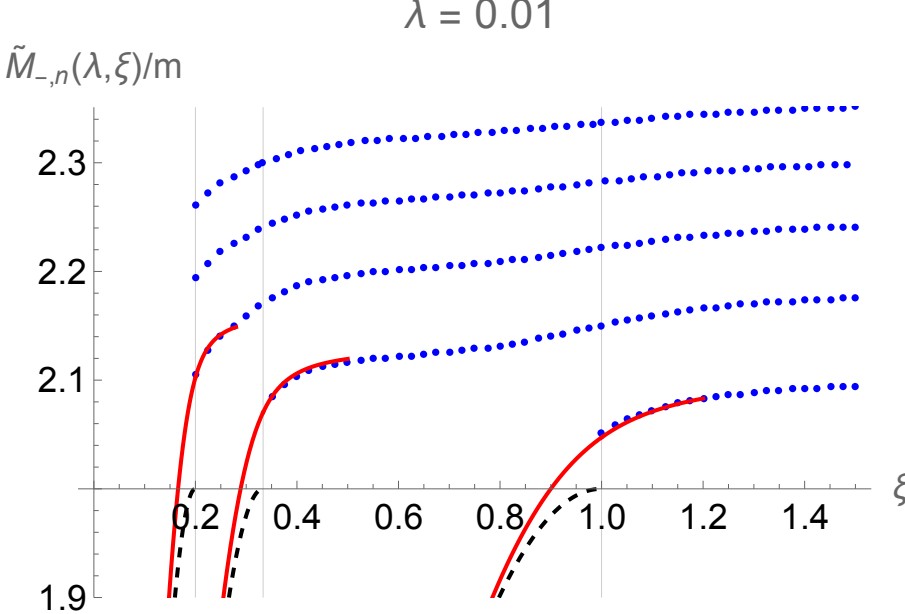

(b) Dashed black lines display the masses of breathers $m_j^{(b)}(\xi)$ with $j = 1, 3, 5$ in the unperturbed sine-Gordon model.

Figure 7: Variation with $\xi$ of the masses of lightest mesons $\tilde{M}_{\iota,n}(\xi)$ with $n = 1, \ldots, 5$ at $\lambda = 0.01$: (a) at $\iota = +$, (b) at $\iota = -$. Blue dots display the semiclassical mass spectra (47), (48). Solid red lines show the masses of the $n = 1$ mesons calculated by means of the modified low-energy expansions: (a) by formula (81) for the mesons with $\iota = +$; (b) by formula (76) for the mesons with $\iota = -$. The masses $m_j^{(b)}(\xi)$ of the breathers in the unperturbed sine-Gordon model, which are shown by black dashed lines, are given by formula (9). All masses are normalised by the soliton mass $m$.

# 6 Conclusions

In this paper we have analytically studied the particle mass spectrum in the double sine-Gordon model in the weak confinement regime, that takes place in this model close to the integrable direction, at which the cosine term with the frequency $\beta/2$ in the scalar self-interaction potential vanishes. This integrability-breaking term induces the weak constant attractive force (string tension) between neighbouring solitons leading to their confinement: two solitons bind into compound particles - the mesons. We obtain three asymptotic expansions in the weak string tension, which describe the meson masses in different regions of the model parameters. Analysis of these asymptotic expansions leads us to the unexpected conclusion, that there is no qualitative difference between the meson and breather excitations in the dsG model in the weak confinement regime. Upon increase of the parameter $\xi$, the $n$th breather does not disappear at the critical point $\xi = 1/n$, as it was in the unperturbed sG model, but transforms into a meson state.

The heuristic perturbative technique, which has been further developed and used in this paper for calculation of the meson masses, originates from ideas due to McCoy and Wu [22]. It is desirable to validate the obtained results by reproducing them in the more systematic and powerful approach based on the Bethe-Salpeter equation. Derivation of the Bethe-Salpeter equation for the dsG field theory would be also crucial for getting insight into the decay mechanism for heavy unstable mesons in this model.

Of course, it would be interesting to perform the numerical study of the particle mass spectra in the dsG model (1) by some direct method [9,10], and to compare the numerical result with our analytical predictions.

# Acknowledgments

I am thankful to Frank Göhmann and Sergei Lukyanov for fruitful discussions. This work was supported by Deutsche Forschungsgemeinschaft (DFG) via Grant BO 3401/7-1.

# A Confinement in the deformed Lieb-Liniger model

The integrable Lieb-Liniger model [31] describes the system of delta-interacting non-relativistic bosons moving in one-dimension. Their state in the $N$-particle sector can be characterized by the wave function $\psi(x_1, x_2, \ldots, x_N)$, which remains unchanged under permutations of the spacial coordinates $x_i, x_j$ of any two particles. In the case of the infinite geometry, the Hamiltonian acting in the $N$-particle sector reads:

$$H_N = -\sum_{j=1}^{N} \frac{\partial_{x_j}^2}{2m} + \sum_{1 \leq i < j \leq N} c\,\delta(x_i - x_j), \tag{93}$$

where $x_i, x_j \in \mathbb{R}$, and $c$ it the interaction constant. The free-boson, and free-fermion cases are realized at $c = 0$, and $1/c = +0$ respectively.

It is well known, that this model admits an alternative equivalent formulation, in which the particle coordinates run in the $N$-dimensional region $\Gamma_N$:

$$\Gamma_N = \{x_1, \ldots, x_N \in \Gamma_N | -\infty < x_1 < x_2 \ldots < x_N < \infty\}. \tag{94}$$

The wave function $\psi(x_1, x_2, \ldots, x_N)$ must satisfy the Robin boundary conditions on $\partial\Gamma_N$:

$$\lim_{x_j \to x_{j+1}} \left[ (\partial_{x_j} - \partial_{x_{j+1}}) + c\,m \right] \psi(x_1, \ldots, x_j, x_{j+1}, \ldots, x_N) = 0, \tag{95}$$

for $j = 1, \ldots, N-1$. The Hamiltonian acting on such wave functions defined in $\Gamma_N$ becomes free:

$$\widetilde{H}_N = -\sum_{j=1}^{N} \frac{\partial_{x_j}^2}{2m}. \tag{96}$$

In the two-particle sector, the solution of the Hamiltonian eigenvalue problem

$$\widetilde{H}_2\, \psi_{p_1,p_2}(x_1, x_1) = \widetilde{E}(p_1, p_2)\, \psi_{p_1,p_2}(x_1, x_1) \tag{97}$$

is given by the Bethe wave function

$$\psi_{p_1,p_2}(x_1, x_1) = \exp[i(p_1 x_1 + p_2 x_2)] + \mathcal{S}(p_1 - p_2) \exp[i(p_1 x_2 + p_2 x_1)], \tag{98}$$

where

$$\mathcal{S}(p_1 - p_2) = \frac{p_1 - p_2 - ic\,m}{p_1 - p_2 + ic\,m} \tag{99}$$

is the two-particle scattering amplitude, and

$$\widetilde{E}(p_1, p_2) = \frac{p_1^2}{2m} + \frac{p_1^2}{2m} \tag{100}$$

is the energy of the state (98).

Besides, a two-particle bound state exists in the attractive regime at $c < 0$. Its wave functions reads

$$\psi_P(x_1, x_2) = \exp\left[ \frac{iP(x_1 + x_2)}{2} + b \cdot (x_1 - x_2) \right], \tag{101}$$

where

$$b = -\frac{c\,m}{2} > 0, \tag{102}$$

and $P$ is the total momentum of this bound state. The energy of the latter equals:

$$\widetilde{E}_1(c, P) = \frac{P^2}{4m} - \frac{mc^2}{4}. \tag{103}$$

Figure 8 illustrates the variation with the parameter $c$ of the energy spectra of the Lieb-Liniger model in the two-particle sector for the states with zero total momentum. The continuous spectrum filling the half-plane $\widetilde{E} > 0$ is shown in grey. The parabola $-mc^2/4$ displays the energy (103) of the two-particle bound state at $P = 0$, which exists in model (93) at $c < 0$.

The model defined by equations (94)-(96) can be viewed as a toy model of a one-dimensional ferromagnet. Indeed, let us introduce the spin operator $\hat{\sigma}(x)$, which acts on the wave-function $\psi(x_1, x_2, \ldots, x_N)$ defined in $\Gamma_N$ as follows:

$$\hat{\sigma}(x)\psi(x_1, x_2, \ldots, x_N) = \sigma(x|x_1, x_2 \ldots, x_N)\, \psi(x_1, x_2, \ldots, x_N). \tag{104}$$

The spin variable $\sigma(x|x_1, x_2 \ldots, x_N)$ takes values $\pm\bar{\sigma}$ for $x$ in the interval $(x_j, x_{j+1})$ bounded by the particle coordinates:

$$\sigma(x|x_1, x_2 \ldots, x_N) = (-1)^j \bar{\sigma}, \text{ for } x \in (x_j, x_{j+1}), \tag{105}$$

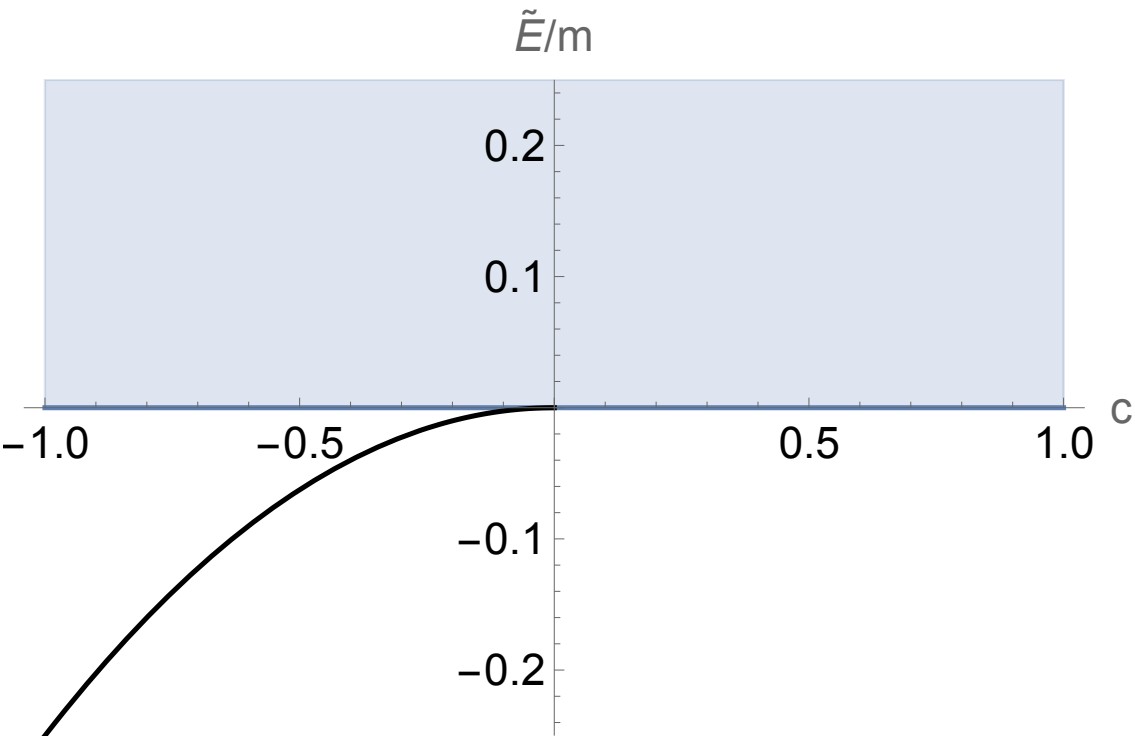

Figure 8: The two-particle energy spectra (100), (103) of model (93) for the states (98), (101) with zero momentum $P = 0$.

with $j = 0, 1, 2, \ldots, N$, and $x_0 = -\infty$, $x_{N+1} = +\infty$. In this interpretation, the particle coordinates $x_j$ are treated as locations of kinks separating the oppositely magnetized ferromagnetic domains.

Let us now deform the Lieb-Liniger model (94)-(96) by application of the uniform magnetic field $h$ coupled to the spin operator $\hat{\sigma}$:

$$\widetilde{H}_N(h) = -\sum_{j=1}^{N} \frac{\partial_{x_j}^2}{2m} - h \int_{-\infty}^{\infty} dx \, [\hat{\sigma}(x) - \bar{\sigma}]. \tag{106}$$

The deformation term breaks integrability of the model and leads to the confinement of kinks. However, in contrast to other QFTs (Ising field theory, Potts field theory, sine-Gordon model, etc.), the deformation term in the Hamiltonian (106) commutes with the operator of number of particles. Furthermore, the model (106) remains integrable in the two-particle sector $N = 2$. The corresponding stationary Schrödinger equation reads:

$$-\frac{\partial_{x_1}^2 \psi(x_1, x_2)}{2m} - \frac{\partial_{x_2}^2 \psi(x_1, x_2)}{2m} + f\,(x_2 - x_1)\,\psi(x_1, x_2) = \widetilde{E}\,\psi(x_1, x_2), \tag{107}$$

where $f = 2h\bar{\sigma}$ is the string tension. The two-particle wave function $\psi(x_1, x_2)$ is defined in the half-plane $-\infty < x_1 < x_2 < \infty$, and satisfies in the line $x_1 = x_2$ the boundary condition:

$$\lim_{x_1 \to x_2} (\partial_{x_1} - \partial_{x_2} + c\,m)\,\psi(x_1, x_2) = 0. \tag{108}$$

For the translation invariant eigenfunction $\psi(x_1, x_2) = \psi(x_1 + X, x_2 + X)$, with an arbitrary real $X$, one obtains from (107) the ordinary differential equation of the Airy type:

$$-\frac{\phi''(x)}{m} - f\,x\,\phi(x) = \widetilde{E}\,\phi(x), \tag{109}$$

where $x = x_1 - x_2 < 0$, and $\phi(x) = \psi(x, 0)$.

The solution of this equation vanishing at $x \to -\infty$ reads:

$$\phi(x) = \text{Ai}\left(-\lambda^{1/3}m\,x - \frac{\widetilde{E}}{\lambda^{2/3}m}\right), \tag{110}$$

where $\lambda = f/m^2$.

The boundary condition

$$\lim_{x \to 0-}[2\phi'(x) + cm\,\phi(x)] = 0, \tag{111}$$

leads to the transcendent secular equation

$$Y_\rho(t_n) = 0, \tag{112}$$

where

$$Y_\rho(t) = \text{Ai}'(-t) - \rho\,\text{Ai}(-t), \tag{113}$$

and

$$\rho = \frac{c}{2\lambda^{1/3}}. \tag{114}$$

Solutions $\{t_n\}_{n=1}^\infty$ of equation (112) determine the energy spectrum $\{\widetilde{E}_n\}_{n=1}^\infty$ of the boundary problem (109), (111):

$$\widetilde{E}_n = m\lambda^{2/3}t_n. \tag{115}$$

The limit $\rho \to +\infty$ corresponds to the free-fermionic case. In this limit, the numbers $-t_n$ approach to the zeroes of the Airy function:

$$\lim_{\rho \to +\infty} t_n = z_n, \quad \text{Ai}(-z_n) = 0.$$

At $\rho = 0$ the free-bosonic case is realized, which is illustrated in Figure 9b. In this case the numbers $-t_n$ coincide with the zeroes of the derivative of the Airy function:

$$\lim_{\rho \to 0} t_n = z_n', \quad \text{Ai}'(-z_n') = 0.$$

The plot of the function $Y_\rho(t)$ at $\rho = -1$ is shown in Figure 9c. The first zero of this function is negative $t_1(-1) = -0.566891$, while all other zeroes remain positive.

Finally, at $\rho \to -\infty$, one gets from equations (114), (115):

$$\widetilde{E}_1 = -\frac{mc^2}{4}, \quad \widetilde{E}_2 = m\lambda^{2/3}z_1, \quad \widetilde{E}_3 = m\lambda^{2/3}z_2, \quad \dots. \tag{116}$$

Figure 10 shows the plots of the meson energies $\widetilde{E}_n(c, \lambda)$ given by equation (115) versus the parameter $c$ at $\lambda = 1$.

The energy spectrum (115) has the following properties.

(i) At any fixed $c \in \mathbb{R}$ and $\lambda > 0$, the energy spectrum is discrete.

(ii) At $\lambda > 0$, all meson energies $\widetilde{E}_n(c, \lambda)$, with $n = 1, 2, \dots$, analytically depend on $c$ at $-\infty < c < \infty$.

(iii) At $\widetilde{E} > 0$, the spectrum becomes more and more dense with decreasing $\lambda$, and transforms into the continuous spectrum at $\lambda = 0$.

(iv) The energy of the lightest meson $\widetilde{E}_1(c, \lambda)$ becomes negative at $c < c_0(\lambda)$, where $c_0(\lambda) = 2\lambda^{1/3}\,\text{Ai}'(0)/\text{Ai}(0) \approx -1.4580\,\lambda^{1/3}$.

(v) At $c < 0$, the lightest meson transforms in the limit $\lambda \to +0$ into the bound state of the unperturbed Lieb-Liniger model (93), and

$$\lim_{\lambda \to +0} \widetilde{E}_1(c, \lambda) = -\frac{m\,c^2}{4}, \text{ at } c < 0.$$

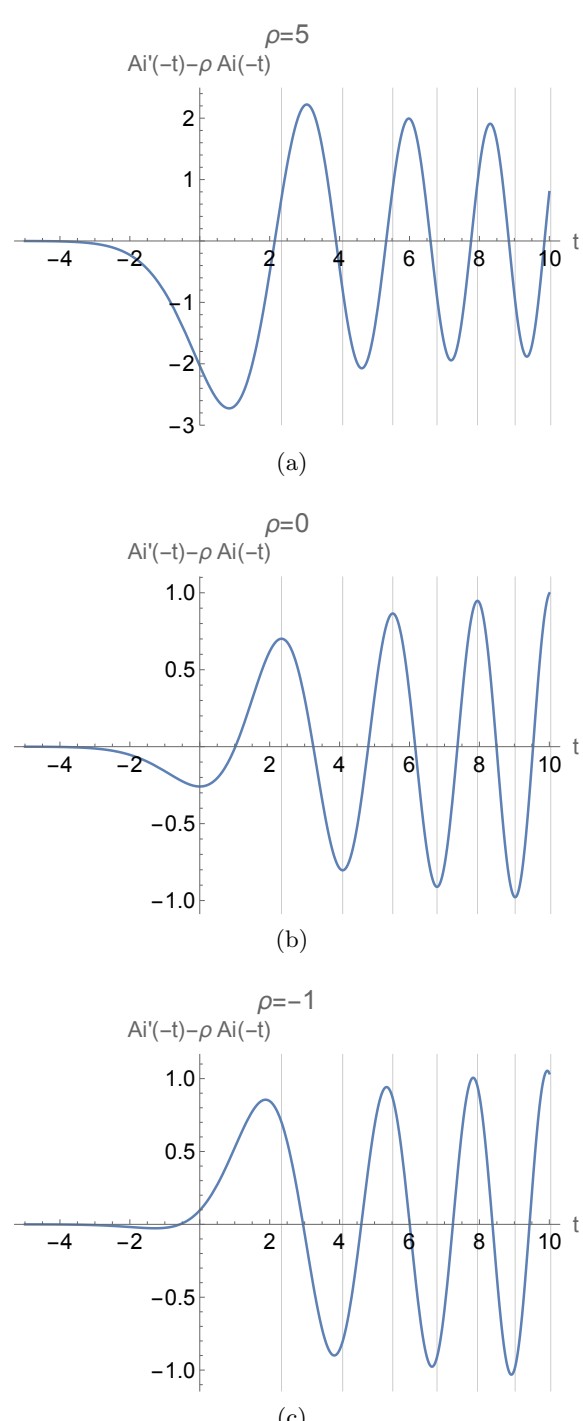

Figure 9: Plots of the function $Y_\rho(t) = \mathrm{Ai}'(-t) - \rho\,\mathrm{Ai}(-t)$ at three different values of the parameter $\rho$: (a) $\rho = 5$; (b) $\rho = 0$; (c) $\rho = -1$. Vertical lines indicate the zeroes $\{z_n\}_{n=1}^6$ of the function $\mathrm{Ai}(-z)$.

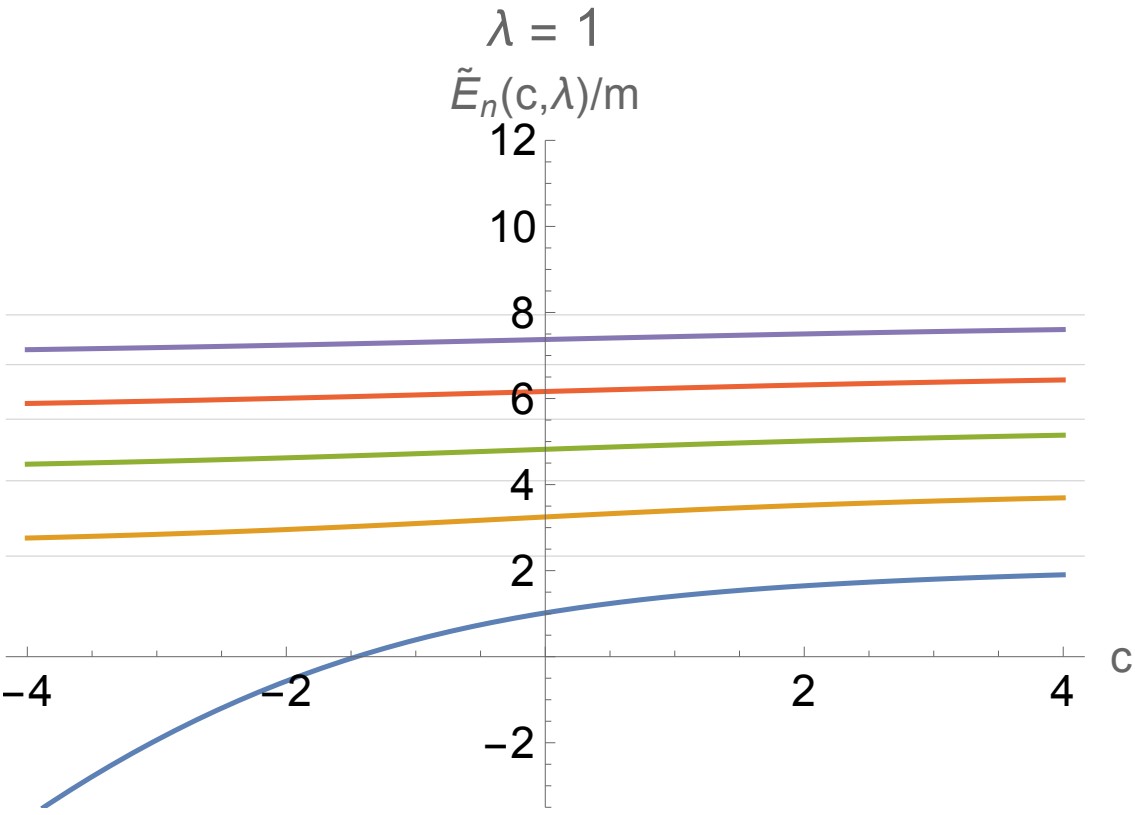

Figure 10: Solid lines display variation with the parameter $c$ of the energies (115) of the mesons with zero momentum $P = 0$ in model (106) at $\lambda = 1$. The dashed black parabolic line shows the energy of the two-particle bound state at $\lambda = 0$. Horizontal lines are located at $\lambda^{2/3} z_n$, with $n = 1 \ldots, 5$.

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
