# Peer review of "Soliton confinement in the double sine-Gordon model"

_SciPost Physics_

## Round 1 · Referee Report · Anonymous (Referee 2) · 2023-12-22

Report
The paper studies the phenomenon of weak confinement in a nonintegrable perturbation of the paradigmatic integrable sine-Gordon field theory. Classically, this theory admits kink solutions that interpolate between adjacent minima of the cosine potential. These kinks become massive particles in the quantum sine-Gordon model. By adding a second cosine perturbation of half the frequency, every other original vacuum state becomes a false vacuum, and the kinks of the original model get confined into bound states called mesons.
Using techniques developed largely by the present author to describe the meson spectrum in other confining theories, the paper derives approximate expressions for the masses of these mesons in the case of weak perturbation. The two main approaches are based on a semiclassical quantisation applicable to higher meson states and on a low-energy expansion to describe the lightest mesons.
However, these techniques need to be modified at and near the so-called reflectionless points where new bound states of kinks (`breathers’) appear in the unperturbed model. One of the novelties of the paper is the construction of the asymptotic expansion near these points by solving first the confinement problem in the Lieb-Liniger model of nonrelativistic bosons with contact interactions. The evolution of the meson spectrum with the coupling is analysed numerically using the obtained asymptotic expansions. It is shown that crossing the reflectionless points, the breathers and mesons are continuously deformed into each other. For example, the breather that would become unstable and would disappear from the spectrum upon increasing the coupling becomes a meson state, so in the perturbed model there is no qualitative difference between these bound states.
Confinement in low dimensional systems has been in the focus of attention in the last years, so the subject of the paper is very timely. The results are valid and interesting. The paper is very well written and accessible. Therefore, I recommend its publication. However, the manuscript could be improved a bit along my comments below which the author may find useful.
Questions and comments:
- Above Eq. (6): the use of the expression “isospin” is a bit misleading, as there is no SU(2) symmetry at play.
- After the nice summary of the bosonic and fermionic cases, perhaps a short comment about the fermionic nature of the results in Eqs. (48,49) as well as about the bosonic nature of the results Eq. (54-57), with reference to Eqs. (16-18) would be helpful.
- Eq. (76): Is it obvious that $S_{reg}$ does not contribute to the scattering length so only $a_0$ appears? (I think its contribution is $\propto\delta\xi$.)
- Eq. (77): It seems to me that the correct equation is
$t_1(\rho) = -\rho^2 + \frac1{2\rho}+\dots$
If this is the case, then Eq. (78) also needs to be slightly modified.
- The $O(\lambda)$ corrections to the breather masses are known (e.g. in Refs. [9,10]). It could be interesting to check that the last term in Eq. (78) agrees with this.
Also, the shifted breather masses could be plotted in Figs. 6 and 7 (dashed lines).
Typos etc.:
- page 2: “it turn out”
- in Eq. (1), shouldn’t the coefficient of the derivative term be 1/(16\pi)?
- below Eq. (3): “model integrability” --> integrability of the model
- several occasions: “remind” --> “remind the reader” or “recall”
- below Eq. (15): “degenerate” --> “become degenerate”
- page 7: “At the first sight” --> “At first sight”
- “account the” --> “account for the”
- “zero order” --> “zeroth order”
- “In the high-energy physics” --> “In high-energy physics”
- “to calculation” --> “to the calculation”
- “It was showed” --> “It was shown”
- page 10: “account this effect” --> “account for this effect”
- page 12: “in the imaginary $\alpha$-axis” --> “on the imaginary $\alpha$-axis”
- “in the right hand side” --> “on the right hand side”
Using techniques developed largely by the present author to describe the meson spectrum in other confining theories, the paper derives approximate expressions for the masses of these mesons in the case of weak perturbation. The two main approaches are based on a semiclassical quantisation applicable to higher meson states and on a low-energy expansion to describe the lightest mesons.
However, these techniques need to be modified at and near the so-called reflectionless points where new bound states of kinks (`breathers’) appear in the unperturbed model. One of the novelties of the paper is the construction of the asymptotic expansion near these points by solving first the confinement problem in the Lieb-Liniger model of nonrelativistic bosons with contact interactions. The evolution of the meson spectrum with the coupling is analysed numerically using the obtained asymptotic expansions. It is shown that crossing the reflectionless points, the breathers and mesons are continuously deformed into each other. For example, the breather that would become unstable and would disappear from the spectrum upon increasing the coupling becomes a meson state, so in the perturbed model there is no qualitative difference between these bound states.
Confinement in low dimensional systems has been in the focus of attention in the last years, so the subject of the paper is very timely. The results are valid and interesting. The paper is very well written and accessible. Therefore, I recommend its publication. However, the manuscript could be improved a bit along my comments below which the author may find useful.
Questions and comments:
- Above Eq. (6): the use of the expression “isospin” is a bit misleading, as there is no SU(2) symmetry at play.
- After the nice summary of the bosonic and fermionic cases, perhaps a short comment about the fermionic nature of the results in Eqs. (48,49) as well as about the bosonic nature of the results Eq. (54-57), with reference to Eqs. (16-18) would be helpful.
- Eq. (76): Is it obvious that $S_{reg}$ does not contribute to the scattering length so only $a_0$ appears? (I think its contribution is $\propto\delta\xi$.)
- Eq. (77): It seems to me that the correct equation is
$t_1(\rho) = -\rho^2 + \frac1{2\rho}+\dots$
If this is the case, then Eq. (78) also needs to be slightly modified.
- The $O(\lambda)$ corrections to the breather masses are known (e.g. in Refs. [9,10]). It could be interesting to check that the last term in Eq. (78) agrees with this.
Also, the shifted breather masses could be plotted in Figs. 6 and 7 (dashed lines).
Typos etc.:
- page 2: “it turn out”
- in Eq. (1), shouldn’t the coefficient of the derivative term be 1/(16\pi)?
- below Eq. (3): “model integrability” --> integrability of the model
- several occasions: “remind” --> “remind the reader” or “recall”
- below Eq. (15): “degenerate” --> “become degenerate”
- page 7: “At the first sight” --> “At first sight”
- “account the” --> “account for the”
- “zero order” --> “zeroth order”
- “In the high-energy physics” --> “In high-energy physics”
- “to calculation” --> “to the calculation”
- “It was showed” --> “It was shown”
- page 10: “account this effect” --> “account for this effect”
- page 12: “in the imaginary $\alpha$-axis” --> “on the imaginary $\alpha$-axis”
- “in the right hand side” --> “on the right hand side”
Report
Dear Editor,
I have read through the article
``Soliton confinement in the double sine-Gordon mode'' by Sergei Rudkevich.
I found it to be very interesting and well written.
It contains a lot of valuable predictions for the spectrum of mesons
in the double sine-Gordon model.
These results will no doubt be useful for those studying that model.
I would recommend that the article be published without any hesitation.
Best wishes,
Sergei Lukyanov
I have read through the article
``Soliton confinement in the double sine-Gordon mode'' by Sergei Rudkevich.
I found it to be very interesting and well written.
It contains a lot of valuable predictions for the spectrum of mesons
in the double sine-Gordon model.
These results will no doubt be useful for those studying that model.
I would recommend that the article be published without any hesitation.
Best wishes,
Sergei Lukyanov

---

## Editorial Decision

resubmitted